# APPROXIMATING PARETO FRONTIER THROUGH BAYESIAN-OPTIMIZATION-DIRECTED ROBUST MULTI-OBJECTIVE REINFORCEMENT LEARNING

## ABSTRACT

Many real-word decision or control problems involve multiple conflicting objectives and uncertainties, which requires learned policies are not only Pareto optimal but also robust. In this paper, we proposed a novel algorithm to approximate a representation for robust Pareto frontier through Bayesian-optimization-directed robust multi-objective reinforcement learning (BRMORL). Firstly, environmental uncertainty is modeled as an adversarial agent over the entire space of preferences by incorporating zero-sum game into multi-objective reinforcement learning (MORL). Secondly, a comprehensive metric based on hypervolume and information entropy is presented to evaluate convergence, diversity and evenness of the distribution for Pareto solutions. Thirdly, the agent's learning process is regarded as a black-box, and the comprehensive metric we proposed is computed after each episode of training, then a Bayesian optimization (BO) algorithm is adopted to guide the agent to evolve towards improving the quality of the approximated Pareto frontier. Finally, we demonstrate the effectiveness of proposed approach on challenging multi-objective tasks across four environments, and show our scheme can produce robust policies under environmental uncertainty.

## 1 INTRODUCTION

Reinforcement learning (RL) algorithm has demonstrated its worth in a series of challenging sequential decision making and control tasks, which train policies to optimize a single scalar reward function (Mnih et al., 2015; Silver et al., 2016; Haarnoja et al., 2018; Hwangbo et al., 2019). However, many real-world tasks are characterized by multiple competing objectives whose relative importance (preferences) is ambiguous in most cases. Moreover, uncertainty or perturbation caused by environment dynamic change, is inevitable in real-world scenarios, which may result in lowered agent performance (Pinto et al., 2017; Ji et al., 2018). For instance, autonomous electric vehicle requires trading off transport efficiency and electricity consumption while considering environmental uncertainty (e.g., vehicle mass, tire pressure and road conditions might vary over time). Consider a decision-making problem for traffic mode, as shown in Figure 1. A practitioner or a rule is responsible for picking the appropriate preference among time and cost, and the agent need to determine different policies depending on the chosen trade-off between these two metrics. Whereas, the environment contain uncertainty factors related to actions of other agents or to dynamic changes of Nature, which may lead to more randomness in these two metrics, and makes multi-objective decision-making or control more challenging. If weather factors are taken into account, e.g., heavy rain may cause traffic congestion, which can increase the time and cost of the plan-A, but it not have a significant impact on the two metrics of the plan-B. From this perspective, selecting plan-B is more robust, i.e., a policy is said to be robust if its capability to obtain utility is relatively stable under environmental changes. Therefore, preference and uncertainty jointly affect the decision-making behavior of the agent.

In traditional multi-objective reinforcement learning (MORL), one popular way is scalarization, which is to convert the multi-objective reward vector into a single scalar reward through various techniques (e.g., by taking a convex combination), and then adopt standard RL algorithms to optimize this scalar reward (Vamplew et al., 2011). Unfortunately, it is very tricky to determine an appropriate scalarization, because often common approach only learn an 'average' policy over the space of preferences (Yang et al., 2019), or though the obtained policies can be relatively quickly

Figure 1: Diagram of decision-making problem for traffic mode. If time is crucial, the agent tend to choose plan-A that takes less time, but it costs more. On the other hand, if cost is more important matters, the agent will be inclined to select plan-B that requires less cost, but it takes more time.

adapted to different preferences between performance objectives but are not necessarily optimal. Furthermore, these methods almost did not take into account the robustness of the policies under different preferences, which means the agent cannot learn robust Pareto optimal policies.

In this work, we propose a novel approach to approximate well-distributed robust Pareto frontier through BRMORL. This allows our trained single network model to produce the robust Pareto optimal policy for any specified preference, i.e., the learned policy is not only robust to uncertainty (e.g., random disturbance and environmental change) but also Pareto optimal under different preference conditions. Our algorithm is based on three key ideas, which are also the main contributions of this paper: (1) present a generalized robust MORL framework through modelling uncertainty as an adversarial agent; (2) inspired by Shannon-Wiener diversity index, a novel metric is presented to evaluate diversity and evenness of distribution for Pareto solutions. In addition, combined with hypervolume indicator, a comprehensive metric is designed, which can evaluate the convergence, diversity and evenness for the solutions on the approximated Pareto frontier; (3) regard agent's learning process in each episode as a black-box, and BO algorithm is used to guide agent to evolve towards improving the quality of the Pareto set. Finally, we demonstrate our proposed algorithm outperform competitive baselines on multi-objective tasks across several MuJoCo (Todorov et al., 2012) environments and SUMO (Simulation of Urban Mobility) (Lopez et al., 2018), and show our approach can produce robust policies under environmental uncertainty.

## 2 RELATED WORK

### 2.1 MULTI-OBJECTIVE REINFORCEMENT LEARNING

MORL algorithms can be roughly classified into two main categories: single-policy approaches and multiple-policy approaches (Roijers et al., 2013; Liu et al., 2014). Single-policy methods seek to find the optimal policy for a given preference among multiple competing objectives. These approaches convert the multi-objective problem into a single-objective problem through different forms of scalarization, including linear and non-linear ones (Mannor & Shimkin, 2002; Tesauro et al., 2008). The main advantage of scalarization is its simplicity, which can be integrated into single-policy scheme with very little modification. However, the main drawback of these approaches is that the preference among the objectives must be set in advance.

Multi-policy methods aim to learn a set of policies that approximate Pareto frontier under different preference conditions. The most common approaches repeatedly call a single-policy scheme with different preferences (Natarajan & Tadepalli, 2005; Van Moffaert et al., 2013; Zuluaga et al., 2016). Other methods learn a set of policies simultaneously via using a multi-objective extended version of value-based RL (Barrett & Narayanan, 2008; Castelletti et al., 2012; Van Moffaert & Nowé, 2014; Mossalam et al., 2016; Nottingham et al., 2019) or via modifying policy-based RL as a MORL variant (Pirotta et al., 2015; Parisi et al., 2017; Abdolmaleki et al., 2020; Xu et al., 2020). Nevertheless, most of these methods are offen constrained to convex regions of the Pareto front and explicitly maintain sets of policies, which may prevent these schemes from finding the sets of well-distributed Pareto solutions which can represent different preferences. There are also meta-policy methods, which can be relatively quickly adapted to different preferences (Chen et al., 2018; Abels et al., 2019; Yang et al., 2019). Although the above works were successful to some extent, these approaches share the same shortcomings that no attention is paid to the robustness of Pareto-optimal policy over the

entire space of preferences. In addition, most approaches still focus on the domains with discrete action space. In contrast, our scheme can guarantee the learned policies is approximately robust Pareto-optimal on continuous control tasks.

## 2.2 Robust reinforcement learning

Robust reinforcement learning (RRL) algorithms can be broadly grouped into three distinct methods (Derman et al., 2020). The first approach focuses on solving robust Markov decision process (MDP) with rectangular uncertainty sets. Some researches proposed RRL algorithms for learning optimal policies using coupled uncertainty sets (Mannor et al., 2012). Other works modeled an ambiguous linear function of a factor matrix as a selection setting from an uncertainty set (Goyal & Grand-Clement, 2018). The second RRL approach considered a distribution over the uncertainty set to mitigate the conservativeness. Yu & Xu (2015) presented the distributional RRL method by supposing the uncertain parameters are random variables following an unknown distribution. Tirinzoni et al. (2018) proposed a RRL scheme using conditioned probability distribution that defines uncertainty sets. A third RRL method mostly concerns adversarial setting in RL. Pinto et al. (2017) developed a robust adversarial reinforcement learning (RARL) scheme through modeling uncertainties via adversarial agent which applies disturbances to the system. Tessler et al. (2019) proposed an adversarial RRL framework through structuring probabilistic action robust MDP and noisy action robust MDP. Nonetheless, these researches do not take into account the connection between Pareto-optimal policy and robust policy, which leaves room for improving the performance of them in practical applications. In contrast, our scheme can learn robust Pareto-optimal policies through modeling uncertainty as an adversary over the entire space of preferences.

## 3 Background

### 3.1 Multi-objective Markov decision process

In this work, we consider a MORL problem defined by a multi-objective Markov decision process (MOMDP), which is represented by the tuple $\langle \mathcal{S}, \mathcal{A}, \mathcal{P}, \boldsymbol{R}, \gamma, \Omega, U_{\boldsymbol{\Omega}} \rangle$ with state space $S$, action space $A$, state transition probability $\mathcal{P}(s'|s, a)$, vector reward function $\boldsymbol{R}(s, a) = [r_1, ..., r_k]^{\mathrm{T}}$, the space of preferences $\Omega$, and preference functions, e.g., $U_{\boldsymbol{\omega}}(\boldsymbol{R})$ which produces an utility function using preference $\boldsymbol{\omega} \in \Omega$, and a discount factor $\gamma \in [1, 0)$. In MOMDP, a policy $\pi$ is associated with a vector of expected returns $\boldsymbol{Q}^{\pi}(s, a) = [Q_1^{\pi}, ..., Q_k^{\pi}]^{\mathrm{T}}$, where the action-value function of $\pi$ for objective $k$ can be represented as $Q_k^{\pi}(s, a) = \mathbb{E}^{\pi}[\sum_t \gamma^t r_k(s_t, a_t)|s_0 = s, a_0 = a]$. For MOMDP, a set of non-dominated policies is called as the Pareto frontier.

**Definition 1.** *A policy $\pi_1$ Pareto dominates another policy $\pi_2$, i.e., $\pi_1 \succ \pi_2$ when*

$$\exists i : Q_i^{\pi_1}(s, a) > Q_i^{\pi_2}(s, a) \land \forall j \neq i : Q_j^{\pi_1}(s, a) \geqslant Q_j^{\pi_2}(s, a).$$

**Definition 2.** *A policy $\pi$ is Pareto optimal if and only if it is non-dominated by any other policies.*

### 3.2 Two-person zero-sum games

In standard two-person zero-sum games, players have opposite goals—the payoff of a player equals the loss of the opponent (Mazalov, 2014), i.e., $\mathcal{V} + \bar{\mathcal{V}} = 0$, where $\mathcal{V}$ and $\bar{\mathcal{V}}$ are payoff of a player and the opponent, respectively.

For two player discounted zero-sum Markov game, assuming protagonist is playing policy $\pi$ and adversary is playing the policy $\bar{\pi}$, transition kernel $\mathcal{P}(s'|s, a, \bar{a})$ depend on both players. In the game, the value function based on $\pi$ and $\bar{\pi}$ can be represented as $v^{\pi, \bar{\pi}}(\mathbf{s}) \equiv \mathbb{E}^{\pi, \bar{\pi}}[\sum_{t=0}^{\infty} \gamma^t r(s_t, a_t, \bar{a}_t) \mid s_0 = s], \forall s \in \mathcal{S}$. Each player chooses his policy regardless of the opponent. Protagonist attempts to maximize the value function (i.e., total expected discounted reward), and adversary seeks to minimize this function.

Nash equilibrium is a key role in game theory, which is one kind of game solution concept. A Nash equilibrium $(\pi^*, \bar{\pi}^*)$ in zero-sum Markov game exists when the following relation holds (Shapley,

1953; Başar & Olsder, 1998):

$$v^*(\mathbf{s}) = \max_{\pi} \min_{\bar{\pi}} \mathbb{E}^{\pi,\bar{\pi}}[\sum_{t=0}^{\infty} \gamma^t r(s_t, a_t, \bar{a}_t) \mid s_0 = s] \tag{1}$$

$$= \min_{\bar{\pi}} \max_{\pi} \mathbb{E}^{\pi,\bar{\pi}}[\sum_{t=0}^{\infty} \gamma^t r(s_t, a_t, \bar{a}_t) \mid s_0 = s], \tag{2}$$

where $\pi^*$ and $\bar{\pi}^*$ are the optimal policies of protagonist and adversary respectively, $v^*$ is optimal equilibrium value of the game. In such a situation, neither player can improve their respective returns, and there is an important relation., i.e., $\forall \pi, \bar{\pi}, v^{\pi,\bar{\pi}^*} \leq v^* \leq v^{\pi^*,\bar{\pi}}$.

## 4 BAYESIAN-OPTIMIZATION-DIRECTED ROBUST MORL

### 4.1 OVERVIEW

We propose a generalized robust MORL framework to learn a single parametric representation for robust Pareto optimal policy over the space of preferences (see Algorithm 1 for implementation scheme based on DDPG). The optimization process of our proposed approach is illustrated in Figure 2. Bayesian model based on Gaussian process is adopted to predict the Pareto quality and estimate the model uncertainty. Then, using the Bayesian model, acquisition function (Frazier, 2018) can determine optimal guess point, which is the suggested preference in our task. In order to prevent the policy from falling into local optimum, some preferences is randomly sampled from replay buffer, which guide the training of the agent together with the preferences from BO. In addition, the policy of the adversary evolves in the opposite direction to the policy of the protagonist in each preference.

In Sections 4.2 and 4.3, through incorporating zero-sum game into MORL, environmental uncertainty is modeled as an adversarial agent. This means that the protagonist needs to learn Pareto optimal policy under attack from the adversary. In Section 4.4, inspired by Shannon-Wiener diversity index, a novel metric for Pareto quality is presented to evaluate the distribution of Pareto solutions from diversity and evenness. Moreover, combined with hypervolume index, a comprehensive metric is designed, which can evaluate the convergence, diversity and evenness for solutions in Pareto set. In Section 4.5, regard agent's learning process as a black-box, and the comprehensive metric for the approximated Pareto frontier is computed after each episode of training, then BO algorithm is adopted to guide the protagonist to evolve towards improving the Pareto quality (i.e., maximizing the comprehensive metric).

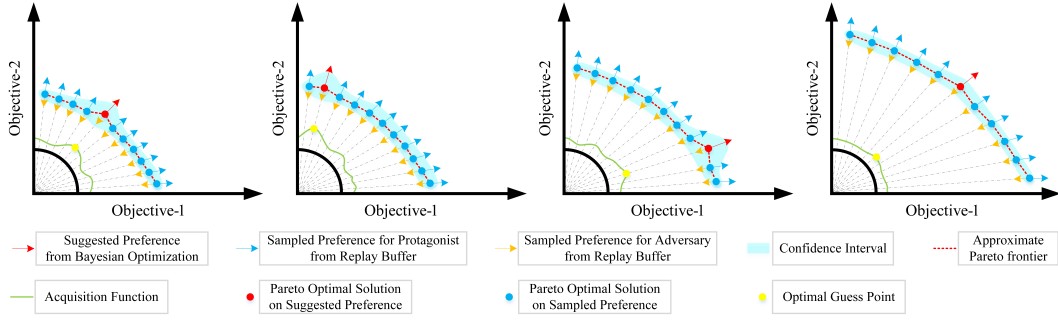

Figure 2: Illustration for process to approximate well-distributed robust Pareto frontier through the proposed algorithm.

### 4.2 ROBUST MULTI-OBJECTIVE MDP

In this section, we propose a robust multi-objective MDP (RMO-MDP), which considers both the Pareto optimality and robustness for the learned policies. Probabilistic action robust MDP (PR-MDP) (Tessler et al., 2019) is adopted to improve the robustness of the policies, which can be

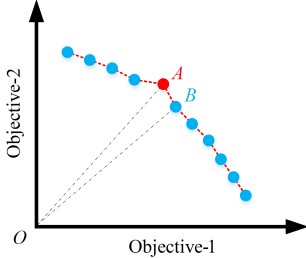
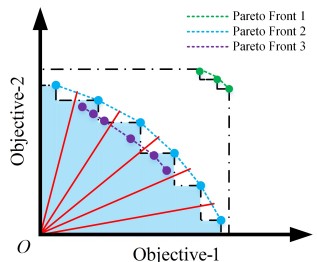

Figure 3: Illustration of the mismatch between the Pareto optimal solution and the corresponding preference. Suppose the point $A$ represents a Pareto optimal solution, which and the origin form the vector $\vec{OA}$. The corresponding preference vector can be represented by the vector $\vec{OB}$. In most cases, $\vec{OA}$ is not parallel to $\vec{OB}$.

Figure 4: Quality analysis of Pareto frontiers. The Pareto frontiers 1, 2 and 3 are approximated by different approaches. The green, blue and purple points represent the solutions on Pareto frontiers 1, 2 and 3 respectively. The hypervolume formed by the solutions on Pareto front 2 and the reference point $O$ is the blue shaded region.

regarded as a special zero-sum game between a protagonist and an adversary. We refer to the optimal policies of the protagonist as robust Pareto-optimal policies in RMO-MDP, which the difference from the MOMDP is that the action space here includes not only the actions of the protagonist, but also the actions of the adversary with a certain probability.

**Definition 3.** *A RMO-MDP can be defined by the tuple $\langle \mathcal{S}, \mathcal{A}^{\mathrm{mix}}, \mathcal{P}, \boldsymbol{R}, \gamma, \Omega, U_{\boldsymbol{\Omega}} \rangle$. $\mathcal{A}^{\mathrm{mix}}$ is the mixed action space. The mixed policy $\pi_{\alpha}^{\mathrm{mix}}(\pi, \bar{\pi})$ is defined as $\pi_{\alpha}^{\mathrm{mix}}(a^{\mathrm{mix}} \mid s, \boldsymbol{\omega}) \equiv (1 - \alpha)\pi(a \mid s, \boldsymbol{\omega}) + \alpha\bar{\pi}(\bar{a} \mid s, \boldsymbol{\omega}), \forall s \in \mathcal{S}$ and $\alpha \in [0, 1]$. $\pi$ and $\bar{\pi}$ are policies the players can take, and $a^{\mathrm{mix}} \sim \pi_{\alpha}^{\mathrm{mix}}(\pi(s), \bar{\pi}(s))$.*

In this work, in order to improve the quality of the approximated Pareto frontier, the scalar utility function $U_{\boldsymbol{\Omega}}$ is designed as non-linear combinations of objectives:

$$U_{\boldsymbol{\Omega}}(s, a^{\mathrm{mix}}, \boldsymbol{\omega}) = \boldsymbol{\omega}^{\mathsf{T}} \boldsymbol{Q}^{\pi_{\alpha}^{\mathrm{mix}}(\pi, \bar{\pi})}(s, a^{\mathrm{mix}}, \boldsymbol{\omega}) + kM(s, a^{\mathrm{mix}}, \boldsymbol{\omega}), \tag{3}$$

$$M(s, a^{\mathrm{mix}}, \boldsymbol{\omega}) = \left\| \frac{\boldsymbol{Q}(s, a^{\mathrm{mix}}, \boldsymbol{\omega})}{\|\boldsymbol{Q}(s, a^{\mathrm{mix}}, \boldsymbol{\omega})\|_2} - \frac{\boldsymbol{\omega}}{\|\boldsymbol{\omega}\|_2} \right\|_2^2, \tag{4}$$

$$\boldsymbol{Q}^{\pi_{\alpha}^{\mathrm{mix}}(\pi, \bar{\pi})}(s, a^{\mathrm{mix}}, \boldsymbol{\omega}) = (1 - \alpha)\boldsymbol{Q}(s, a, \boldsymbol{\omega}) + \alpha\boldsymbol{Q}(s, \bar{a}, \boldsymbol{\omega}), \tag{5}$$

where $M(s, a^{\mathrm{mix}}, \boldsymbol{\omega})$ is a metric, which can evaluate the mismatch between the Pareto optimal solution and the corresponding preference. Figure 3 illustrates metric function $M(s, a^{\mathrm{mix}}, \boldsymbol{\omega})$ in more detail. The distribution of solutions on the Pareto front can be more well-distributed through optimizing function $M(s, a^{\mathrm{mix}}, \boldsymbol{\omega})$. $k$ is a coefficient can adjust the role of $M(s, a^{\mathrm{mix}}, \boldsymbol{\omega})$ in the utility function 3. For a protagonist, $k$ is a negative, and $k$ is positive for an adversary. This means that the policy with higher preference is more likely to be violently attacked by an adversary, which can makes the policy with higher preference stronger robust.

Under the condition of adversary attack, the utility value of protagonist's policy can be defined as $v_{\alpha}^{\pi} \equiv \min_{\bar{\pi}} \mathbb{E}^{\pi_{\alpha}^{\mathrm{mix}}(\pi, \bar{\pi})}[U_{\boldsymbol{\Omega}}(s, a^{\mathrm{mix}}, \boldsymbol{\omega})]$. Therefore, the robust Pareto optimal policy is optimal policy in RMO-MDP, which can be represent as:

$$\pi_{\alpha}^{*} \in \arg\max_{\pi} \min_{\bar{\pi}} \mathbb{E}^{\pi_{\alpha}^{\mathrm{mix}}(\pi, \bar{\pi})}[U_{\boldsymbol{\Omega}}(s, a^{\mathrm{mix}}, \boldsymbol{\omega})]. \tag{6}$$

The complexity of greedy solution to finding the Nash equilibria policies is exponential in the cardinality of the action space, which makes it unworkable in most cases (Schulman et al., 2015). In addition, most two player discounted zero-sum Markov game methods require solving for the equilibrium policy of a minimax action-value function at each iteration. This is a typically intractable optimization problem (Pinto et al., 2017). Instead, we focus on approximating equilibrium solution to avoid this tricky optimization.

### 4.3 POLICY ITERATION FOR RMO-MDP

In this section, we present a policy iteration (PI) approach for solving RMO-MDP called robust multi-objective PI (RMO-PI). RMO-PI algorithm can decompose the RMO-MDP problem into two sub-problems (policy evaluation and policy improvement) and iterate until convergence.

#### 4.3.1 ROBUST MULTI-OBJECTIVE POLICY EVALUATION

In this stage, the vectorized $\boldsymbol{Q}$-function is learned to evaluate the policy $\pi$ of the protagonist. With Equation 5, we define the target vectorized $\boldsymbol{Q}$-function as:

$$
\begin{aligned}
\boldsymbol{y} &= \mathbb{E}^{\pi_\alpha^{\mathrm{mix}}}[\boldsymbol{R} + \gamma \boldsymbol{Q}^{\pi_\alpha^{\mathrm{mix}}(\pi,\bar{\pi})}(s', a^{\mathrm{mix}}, \boldsymbol{\omega}; \phi^-)] \\
&= \mathbb{E}_{s',a^{\mathrm{mix}},\boldsymbol{\omega}}\left\{\boldsymbol{R} + \gamma[(1-\alpha)\boldsymbol{Q}(s', a, \boldsymbol{\omega}; \phi^-) + \alpha\boldsymbol{Q}(s', \bar{a}, \boldsymbol{\omega}; \phi^-)]\right\},
\end{aligned}
\tag{7}
$$

Then, we minimize the following loss function at each step:

$$
L_1(\phi) = \mathbb{E}^{\pi_\alpha^{\mathrm{mix}}}\left[\|\boldsymbol{y}_{\boldsymbol{\omega}_{rb}} - \boldsymbol{Q}(s, a, \boldsymbol{\omega}_{rb}; \phi)\|_2^2 + \|\boldsymbol{y}_{\boldsymbol{\omega}_{bo}} - \boldsymbol{Q}(s, a, \boldsymbol{\omega}_{bo}; \phi)\|_2^2\right],
\tag{8}
$$

where $\phi$ and $\phi^-$ are the parameters of the $\boldsymbol{Q}$-function network and the target $\boldsymbol{Q}$-function network, $\boldsymbol{\omega}_{rb}$ and $\boldsymbol{\omega}_{bo}$ are obtained from replay buffer and Bayesian-optimization, $\boldsymbol{y}_{\boldsymbol{\omega}_{rb}}$ and $\boldsymbol{y}_{\boldsymbol{\omega}_{bo}}$ represent $\boldsymbol{y}(s', a^{\mathrm{mix}}, \boldsymbol{\omega}_{rb})$ and $\boldsymbol{y}(s', a^{\mathrm{mix}}, \boldsymbol{\omega}_{bo})$, respectively. In order to improve the smoothness of the landscape of loss function, the auxiliary loss setting is used (Yang et al., 2019):

$$
L_2(\phi) = \mathbb{E}^{\pi_\alpha^{\mathrm{mix}}}\left[\|\boldsymbol{\omega}_{rb}^{\mathsf{T}}\boldsymbol{y}_{\boldsymbol{\omega}_{rb}} - \boldsymbol{\omega}_{rb}^{\mathsf{T}}\boldsymbol{Q}(s, a, \boldsymbol{\omega}_{rb}; \phi)\|_2^2 + \|\boldsymbol{\omega}_{bo}^{\mathsf{T}}\boldsymbol{y}_{\boldsymbol{\omega}_{bo}} - \boldsymbol{\omega}_{bo}^{\mathsf{T}}\boldsymbol{Q}(s, a, \boldsymbol{\omega}_{bo}; \phi)\|_2^2\right].
\tag{9}
$$

The final loss function can be written as: $L(\phi) = (1-\beta)L_1(\phi) + \beta L_2(\phi)$, where $\beta$ is a weighting coefficient to trade off between losses $L_1(\phi)$ and $L_2(\phi)$.

#### 4.3.2 ROBUST MULTI-OBJECTIVE POLICY IMPROVEMENT

In RMO-PI, policy improvement refers to optimizing and updating the policies of a protagonist and an adversary for the given utility function. RMO-PI optimizes both of the agents through the following alternating process. In the first stage, the policy of protagonist is learned while holding the adversary's policy fixed. In the second stage, the policy of protagonist is held constant and the adversary's policy is learned. This learning sequence is repeated until convergence.

The protagonist seeks to maximize the utility function $U_{\boldsymbol{\Omega}}$, and then the policy gradient can be represented as: $\nabla_\theta L^\pi = \nabla_\theta L_{\boldsymbol{\omega}_{rb}}^\pi + \nabla_\theta L_{\boldsymbol{\omega}_{bo}}^\pi$, where

$$
\nabla_\theta L_{\boldsymbol{\omega}_{rb}}^\pi \approx \mathbb{E}^{\pi_\alpha^{\mathrm{mix}}}[(1-\alpha)\nabla_a \boldsymbol{\omega}_{rb}^{\mathsf{T}}\boldsymbol{Q}(s, a, \boldsymbol{\omega}_{rb}; \phi)\nabla_\theta \pi(s, \boldsymbol{\omega}_{rb}; \theta) + k\nabla_a M(s, a, \boldsymbol{\omega}_{rb})\nabla_\theta \pi(s, \boldsymbol{\omega}_{rb}; \theta)],
\tag{10}
$$

$$
\nabla_\theta L_{\boldsymbol{\omega}_{bo}}^\pi \approx \mathbb{E}^{\pi_\alpha^{\mathrm{mix}}}[(1-\alpha)\nabla_a \boldsymbol{\omega}_{bo}^{\mathsf{T}}\boldsymbol{Q}(s, a, \boldsymbol{\omega}_{bo}; \phi)\nabla_\theta \pi(s, \boldsymbol{\omega}_{bo}; \theta) + k\nabla_a M(s, a, \boldsymbol{\omega}_{bo})\nabla_\theta \pi(s, \boldsymbol{\omega}_{bo}; \theta)],
\tag{11}
$$

$\theta$ is the model parameters of the protagonist.

Next, the adversary tries to minimize the utility function $U_{\boldsymbol{\Omega}}$, and the policy gradient can be written as: $\nabla_{\bar{\theta}} L^{\bar{\pi}} = \nabla_{\bar{\theta}} L_{\boldsymbol{\omega}_{rb}}^{\bar{\pi}} + \nabla_{\bar{\theta}} L_{\boldsymbol{\omega}_{bo}}^{\bar{\pi}}$, where

$$
\nabla_{\bar{\theta}} L_{\boldsymbol{\omega}_{rb}}^{\bar{\pi}} \approx \mathbb{E}^{\pi_\alpha^{\mathrm{mix}}}[\alpha\nabla_{\bar{a}} \boldsymbol{\omega}_{rb}^{\mathsf{T}}\boldsymbol{Q}(s, \bar{a}, \boldsymbol{\omega}_{rb}; \phi)\nabla_{\bar{\theta}} \bar{\pi}(s, \boldsymbol{\omega}_{rb}; \bar{\theta}) + k\nabla_{\bar{a}} M(s, \bar{a}, \boldsymbol{\omega}_{rb})\nabla_{\bar{\theta}} \bar{\pi}(s, \boldsymbol{\omega}_{rb}; \bar{\theta})],
\tag{12}
$$

$$
\nabla_{\bar{\theta}} L_{\boldsymbol{\omega}_{bo}}^{\bar{\pi}} \approx \mathbb{E}^{\pi_\alpha^{\mathrm{mix}}}[\alpha\nabla_{\bar{a}} \boldsymbol{\omega}_{bo}^{\mathsf{T}}\boldsymbol{Q}(s, \bar{a}, \boldsymbol{\omega}_{bo}; \phi)\nabla_{\bar{\theta}} \bar{\pi}(s, \boldsymbol{\omega}_{bo}; \bar{\theta}) + k\nabla_{\bar{a}} M(s, \bar{a}, \boldsymbol{\omega}_{bo})\nabla_{\bar{\theta}} \bar{\pi}(s, \boldsymbol{\omega}_{bo}; \bar{\theta})],
\tag{13}
$$

$\bar{\theta}$ is the model parameters of the adversary. The derivation details of the policy gradients are available in Appendix A.1.2.

### 4.4 METRICS FOR PARETO REPRESENTATION

Since the true Pareto set is intractable to obtain in complex problems, the goal of MORL is to find the set of policies that best approximates the optimal Pareto front. Many researchers have reported the works for quality metrics of Pareto front (Cheng et al., 2012; Parisi et al., 2017; Audet et al., 2018).

Hypervolume indicator is widely adopted to evaluate the quality of an approximated Pareto frontier, which can measure the convergence and uniformity for the distribution of Pareto solutions (Zitzler & Thiele, 1999; Xu et al., 2020). From our perspective, this indicator may be difficult to accurately measure the uniformity of the Pareto solution distribution.

As shown in Figure 4, suppose the Pareto frontiers 1, 2 and 3 are obtained by different algorithms, and compared with the Pareto frontiers 2 and 3, although the hypervolume metric formed by the solutions on Pareto frontier 1 and the reference point O is optimal, the distribution of solutions on the frontier 1 is not well-distributed, which makes the valid preferences of the practitioner or the agent to choose is very limited. Moreover, imagine the solutions on Pareto frontier 1 are very close to each other or even overlap into one solution. At this time, if we adopt the metric (integrated hypervolume metric and sparsity metric) proposed in the paper (Xu et al., 2020) to measure the quality of Pareto frontier 1, the result to have high hypervolume and low sparsity is very ideal. However, such Pareto frontier 1 might not satisfy the needs of the practitioner or the agent. In a word, the high quality of the approximated Pareto frontier is expected to have high hypervolume, and the distribution of solutions is well-distributed. Therefore, in this section, we proposed a novel metric for quality of the approximated Pareto frontier through combining hypervolume metric and evenness metric.

Inspired by Shannon-Wiener diversity index, the diversity metric for the solutions of the Pareto frontier can be expressed as $D(P) = -\sum [p_i \ln (p_i)]$, where $P$ represents the solutions of the Pareto frontier, and $p_i$ is the proportion of the number of non-dominated solutions in the corresponding solution interval to the total number of the solutions on Pareto frontier. The expected diversity of Pareto set $D_{max}$ can be defined as $\ln(S_n)$, and $S_n$ is the number of solution intervals. Then, our evenness metric $E(P)$ can be represented as $D(P)/D_{max}$. For example, in Figure 4, $S_n = 6$, and the evenness metrics for the distribution of the solutions on the Pareto frontiers 1, 2 and 3 are approximately equal to 0.37, 1 and 0.56, respectively. Hence, we can get the following two inferences.

**Proposition 1.** *As $E(P)$ and $S_n$ increases, the distribution of solutions in Pareto set becomes denser and more uniform, and the Pareto frontier becomes more continuous.*

**Proposition 2.** *The Pareto frontier is continuous as $E(P) = 1$ and $S_n \to \infty$.*

Combined with the hypervolume indicator $H(P)$, we propose a comprehensive metric $I(P)$ that can measure the convergence, diversity and evenness of the solutions:

$$I(P) = H(P)(1 + \lambda E(P)), \tag{14}$$

where $\lambda$ is a weight coefficient.

### 4.5 BAYESIAN-OPTIMIZATION-DIRECTED PARETO REPRESENTATION IMPROVEMENT

In this Section, in order to further improve the representation of the approximated Pareto frontier, the agent's learning process is regarded as a black-box, and the comprehensive metric $I(P)$ is computed after each episode of training, then a BO algorithm is adopted to guide the protagonist to evolve towards maximizing the proposed metric $I(P)$. As shown in Figure 5, the Pareto representation improvement scheme based on BO-directed is illustrated. The value of the objective function $f(\Omega)$ equals the value of the comprehensive metric $I(P)$, which is obtain after each episode of training. In addition, suggested preferences from BO algorithm and sampled preferences from replay buffer are simultaneously used to guide the learning process, which is to avoid the algorithm into a local optimum. The scheme to guide the learning process with BO has high universality for Pareto quality improvement, and does not require much expert experience in the selection of prediction models.

## 5 EXPERIMENTS

In order to benchmark our proposed scheme, we develop two MORL environments with continuous action space based on SUMO and Swimmer-v2. Moreover, we also adopt HalfCheetah-v2 and Walker2d-v2, which are two MORL domains provided by Xu et al. (2020). The goal of all tasks is to try to optimize the speed of the agent while minimizing energy consumption. The observation and action space settings are shown in Table 1. The more details can be found in Appendix A.2.

| $\mathcal{D}$ → Training Data | **Surrogate Model** | → | **Posterior** | → | **Acquisition Function** | → $U_{cs}(\Omega)$ |
| $\Omega$ → Candidate Set | Gaussian Process | | $\mathbb{P}(f(\Omega)|\mathcal{D})$ | | Expected Improvement | Utility of the Candidate Set |

Figure 5: Pareto representation improvement scheme based on BO algorithm. The surrogate model for the objective function $f(\Omega)$ is typically a Gaussian Process. Posteriors represent the confidence a model has about the function values at a point or set of points. Acquisition function is employed to evaluate the usefulness of optimal guess point corresponding to posterior distribution over $f(\Omega)$. The expected improvement method chosen to design the acquisition function in our scheme.

Table 1: Observation space and action space of the experiment environments.

|  | SUMO | Swimmer-v2 | HalfCheetah-v2 | Walker2d-v2 |
| --- | --- | --- | --- | --- |
| Observation Space | $\mathcal{S} \in \mathbb{R}^{16}$ | $\mathcal{S} \in \mathbb{R}^{8}$ | $\mathcal{S} \in \mathbb{R}^{17}$ | $\mathcal{S} \in \mathbb{R}^{17}$ |
| Action Space | $\mathcal{A} \in \mathbb{R}^{1}$ | $\mathcal{A} \in \mathbb{R}^{2}$ | $\mathcal{A} \in \mathbb{R}^{6}$ | $\mathcal{A} \in \mathbb{R}^{6}$ |

Our algorithm is implemented based on Deep Deterministic Policy Gradient (DDPG) (Lillicrap et al., 2015) framework. In principle, our scheme can be combined with any RL method, regardless of whether it is off-policy or on-policy. Moreover, we implement three baseline methods for comparison and ablation analysis: SMORL represents a MO-DDPG method based on linear scalarization function, which is a linear combination of rewards in the form of a preference; SRMORL is a RMO-DDPG approach using linear scalarization function; RMORL represents a RMO-DDPG approach with the utility function $U_{\Omega}$. BRMORL is a RMO-DDPG scheme combined with the utility function $U_{\Omega}$ and BO algorithm. More details about the algorithms are described in Appendix A.1.1.

Figure 6 and 7 show the learning curves and Pareto frontiers comparison results on SUMO and Swimmer-v2 respectively. Moreover, the results in Table 2 and 3 demonstrate that our proposed BRMORL scheme outperforms all the baseline methods on SUMO and Swimmer-v2 environments in hypervolume and evenness. It can also be found from Figure 7(c) the BRMORL mothod is not only able to find solutions on the convex portions of the Pareto frontier, but also the concave portions.

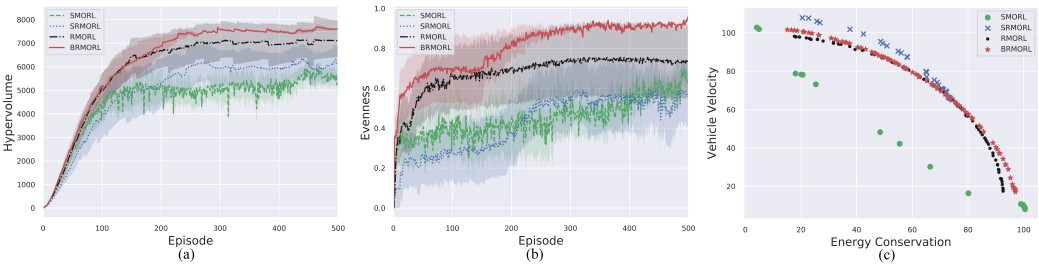

Figure 6: The learning curves and the Pareto frontiers obtained by different algorithms on SUMO.

Table 2: Training results on SUMO.

|  | Hypervolume | Evenness |
| --- | --- | --- |
| SMORL | $4547.43 \pm 1919.14$ | $0.45 \pm 0.30$ |
| SRMORL | $4904.25 \pm 2480.33$ | $0.42 \pm 0.37$ |
| RMORL | $5900.67 \pm 2443.58$ | $0.67 \pm 0.36$ |
| BRMORL | $6219.57 \pm 2164.15$ | $0.81 \pm 0.24$ |

Table 3: Training results on Swimmer-v2.

|  | Hypervolume | Evenness |
| --- | --- | --- |
| SMORL | $3045.63 \pm 2546.65$ | $0.34 \pm 0.33$ |
| SRMORL | $4164.11 \pm 1487.67$ | $0.20 \pm 0.18$ |
| RMORL | $7682.44 \pm 4844.27$ | $0.35 \pm 0.25$ |
| BRMORL | $8118.58 \pm 4344.80$ | $0.74 \pm 0.14$ |

Figure 8 illustrates that the robustness of different policy models under the preference=[0.5,0.5], on Swimmer-v2 domain. We test with jointly varying both mass and disturbance probability. Obviously, the capability to obtain return based on BRMORL approach is less affected by environmental changes than other schemes. Moreover, the standard deviation based on the utility of the policy is adopted to

quantify the robustness. This means that the stronger the robustness of a policy is, then the smaller its standard deviation is. Table 4 shows the quantitative analysis results of robustness under different preferences and environmental changes, on Swimmer-v2. For more results and implementation details, please refer to Appendix A.3 and A.4.1.

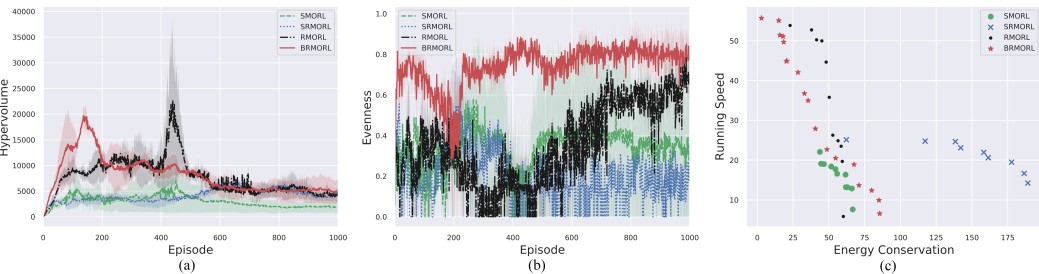

Figure 7: The learning curves and the Pareto frontiers obtained by different methods on Swimmer-v2.

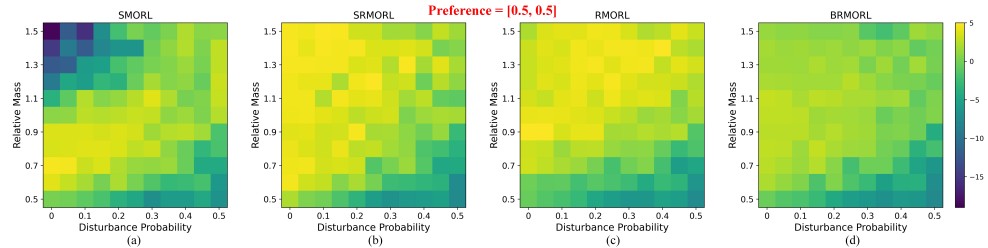

Figure 8: Robustness to environmental uncertainty. Disturbance probability represents the probability of a random disturbance being played instead of the selected action. Relative mass denotes the ratio of the current agent's mass to its original mass.

In Table 5 and 7, we compare our BRMORL scheme with state-of-the-art baseline (PG-MORL) provided by Xu et al. (2020). Although our method is not superior in hypervolume, it outperforms the baseline in evenness, robustness and utility. In this section, the utility is defined as the expectation of return based on a policy under environmental changes. More details and results can be found in Appendix A.3 and A.4.2.

Table 4: Quantitative analysis results for robustness.

|  | [0.1,0.9] | [0.3,0.7] | [0.5,0.5] | [0.7,0.3] | [0.9,0.1] |
|---|---|---|---|---|---|
| SMORL | 8.96 | 9.55 | 4.61 | 9.58 | 16.48 |
| SRMORL | 0.44 | 1.01 | 3.07 | 7.88 | 15.86 |
| RMORL | 1.99 | 0.59 | 3.11 | 8.45 | 14.36 |
| BRMORL | 0.28 | 0.94 | 2.51 | 4.62 | 7.93 |

Table 5: Test results on Walker2d-v2.

|  | PGMORL | BRMORL |
|---|---|---|
| Hypervolume | 57132.70 | 30737.01 |
| Evenness | 0.28 | 0.32 |
| Robustness | 34.91 | 14.53 |
| Utility | -193.86 | -11.63 |

## 6 CONCLUSION AND DISCUSSION

In this paper, we proposed a generalized robust MORL framework to approximate a representation for robust Pareto frontier, which allows our trained single model to produce the robust Pareto optimal policy for any specified preference.

Our experiments across four different domains demonstrate that our scheme is effective and advanced. Most importantly, we note that training with appropriate adversarial setting can not only result in robust policies, but also improve the performance even. Moreover, both solutions on convex and concave portions of the Pareto frontier can be found through our approach. Although our scheme cannot guarantee the learned policy is optimal, it is approximately robust Pareto optimal.

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

# A  APPENDIX

## A.1  ALGORITHM

### A.1.1  ALGORITHM OVERVIEW

The details of the BRMORL, RMORL, SRMORL and SMORL schemes are provided in Algorithm 1, 2, 3 and 4 respectively.

---

**Algorithm 1** Bayesian-optimization-directed robust multi-objective DDPG

---

**Input:** probability $\alpha$, weighting coefficients $\beta, \lambda, \tau, k$
Randomly initialize actor $\pi(s, \boldsymbol{\omega}; \theta)$, adversary $\bar{\pi}(s, \boldsymbol{\omega}; \bar{\theta})$ and critic network $Q(s, a, \boldsymbol{\omega}; \phi)$
Initialize target networks with weights $\theta^-, \bar{\theta}^-$ and $\phi^-$
Initialize replay buffer $B$ and comprehensive metric $I$
**for** $episode = 0...M$ **do**
    Receive initial state $s_0$
    Predict preference $\boldsymbol{\omega}_{bo}$ through Bayesian-optimization
    $\boldsymbol{\omega}_{bo} \leftarrow f_{bo}(I)$
    **for** $t = 0...T$ **do**
        **if** $t \leqslant T/2$ **then**
            $\boldsymbol{\omega}_t \leftarrow \boldsymbol{\omega}_{bo}$
        **else**
            Sample preference $\boldsymbol{\omega}_{ud}$ based on uniform-distribution:
            $\boldsymbol{\omega}_t \leftarrow \boldsymbol{\omega}_{ud}$
        **end if**
        Sample action $a_t = \begin{cases} \pi(s_t, \boldsymbol{\omega}_t; \theta) & \text{, w.p. } (1 - \alpha) \\ \bar{\pi}(s_t, \boldsymbol{\omega}_t; \bar{\theta}) & \text{, otherwise} \end{cases}$
        $\tilde{a}_t = a_t$ + exploration noise
        Execute action $\tilde{a}_t$ and observe reward $r_t$ and new state $s_{t+1}$
        Store transition $(s_t, \boldsymbol{\omega}_t, \tilde{a}_t, r_t, \mathbf{s}_{t+1})$ in $B$
        Extend the dimension of preference $\boldsymbol{\omega}_{bo}$ to batch size
        **for** $i = 0...N$ **do**
            Sample batch from replay buffer $B$
            Update actor according to equations 10 and 11:
            $\theta \leftarrow \nabla_\theta L^\pi_{\boldsymbol{\omega}_{rb}} + \nabla_\theta L^\pi_{\boldsymbol{\omega}_{bo}}$
            Update critic according to equations 8 and 9:
            $\phi \leftarrow (1 - \beta)\nabla_\phi L_1(\phi) + \beta\nabla_\phi L_2(\phi)$
        **end for**
        Sample batch from replay buffer $B$
        Update adversary according to equations 12 and 13:
        $\bar{\theta} \leftarrow \nabla_{\bar{\theta}} L^{\bar{\pi}}_{\boldsymbol{\omega}_{rb}} + \nabla_{\bar{\theta}} L^{\bar{\pi}}_{\boldsymbol{\omega}_{bo}}$
        Update the target networks:
                 $\theta^- \leftarrow \tau\theta + (1 - \tau)\theta^-$
                 $\bar{\theta}^- \leftarrow \tau\bar{\theta} + (1 - \tau)\bar{\theta}^-$
                 $\phi^- \leftarrow \tau\phi + (1 - \tau)\phi^-$
    **end for**
    Compute comprehensive metric $I(P)$ using equation 14 and Pareto solution set $P$
**end for**

---

---

**Algorithm 2** Robust multi-objective DDPG with the utility function $U_{\boldsymbol{\Omega}}$

---

**Input:** probability $\alpha$, weighting coefficients $\beta, \tau, k$
Randomly initialize actor $\pi(s, \boldsymbol{\omega}; \theta)$, adversary $\bar{\pi}(s, \boldsymbol{\omega}; \bar{\theta})$ and critic network $Q(s, a, \boldsymbol{\omega}; \phi)$
Initialize target networks with weights $\theta^-, \bar{\theta}^-$ and $\phi^-$
Initialize replay buffer $B$ and comprehensive metric $I$
**for** $episode = 0...M$ **do**
   Receive initial state $s_0$
   **for** $t = 0...T$ **do**
      Sample preference $\boldsymbol{\omega}_{ud}$ based on uniform-distribution:
      $\boldsymbol{\omega}_t \leftarrow \boldsymbol{\omega}_{ud}$
      Sample action $a_t = \begin{cases} \pi(s_t, \boldsymbol{\omega}_t; \theta) & \text{, w.p. } (1 - \alpha) \\ \bar{\pi}(s_t, \boldsymbol{\omega}_t; \bar{\theta}) & \text{, otherwise} \end{cases}$
      $\tilde{a}_t = a_t +$ exploration noise
      Execute action $\tilde{a}_t$ and observe reward $r_t$ and new state $s_{t+1}$
      Store transition $(s_t, \boldsymbol{\omega}_t, \tilde{a}_t, r_t, \mathbf{s}_{t+1})$ in $B$
      **for** $i = 0...N$ **do**
         Sample batch from replay buffer $B$
         Update actor according to equations 10:
         $\theta \leftarrow \nabla_\theta L^\pi_{\boldsymbol{\omega}_{rb}}$
         Update critic network:
         $\phi \leftarrow (1 - \beta) \nabla_\phi \mathbb{E}^{\pi^{\text{mix}}_\alpha} \left[ \|\boldsymbol{y}_{\boldsymbol{\omega}_{rb}} - \boldsymbol{Q}(s, a, \boldsymbol{\omega}_{rb}; \phi)\|^2_2 \right] +$
         $\beta \nabla_\phi \mathbb{E}^{\pi^{\text{mix}}_\alpha} \left[ \|\boldsymbol{\omega}^{\mathsf{T}}_{rb} \boldsymbol{y}_{\boldsymbol{\omega}_{rb}} - \boldsymbol{\omega}^{\mathsf{T}}_{rb} \boldsymbol{Q}(s, a, \boldsymbol{\omega}_{rb}; \phi)\|^2_2 \right]$
      **end for**
      Sample batch from replay buffer $B$
      Update adversary according to equations 12:
      $\bar{\theta} \leftarrow \nabla_{\bar{\theta}} L^{\bar{\pi}}_{\boldsymbol{\omega}_{rb}}$
      Update the target networks:
         $\theta^- \leftarrow \tau\theta + (1 - \tau)\theta^-$
         $\bar{\theta}^- \leftarrow \tau\bar{\theta} + (1 - \tau)\bar{\theta}^-$
         $\phi^- \leftarrow \tau\phi + (1 - \tau)\phi^-$
   **end for**
**end for**

---

---

**Algorithm 3** Robust multi-objective DDPG with linear scalarization function

---

**Input:** probability $\alpha$, weighting coefficients $\beta, \tau$
Randomly initialize actor $\pi(s, \boldsymbol{\omega}; \theta)$, adversary $\bar{\pi}(s, \boldsymbol{\omega}; \bar{\theta})$ and critic network $Q(s, a, \boldsymbol{\omega}; \phi)$
Initialize target networks with weights $\theta^-, \bar{\theta}^-$ and $\phi^-$
Initialize replay buffer $B$ and comprehensive metric $I$
**for** $episode = 0...M$ **do**
   Receive initial state $s_0$
   **for** $t = 0...T$ **do**
      Sample preference $\boldsymbol{\omega}_{ud}$ based on uniform-distribution:
      $\boldsymbol{\omega}_t \leftarrow \boldsymbol{\omega}_{ud}$
      Sample action $a_t = \begin{cases} \pi(s_t, \boldsymbol{\omega}_t; \theta) & \text{, w.p. } (1 - \alpha) \\ \bar{\pi}(s_t, \boldsymbol{\omega}_t; \bar{\theta}) & \text{, otherwise} \end{cases}$
      $\tilde{a}_t = a_t$ + exploration noise
      Execute action $\tilde{a}_t$ and observe reward $r_t$ and new state $s_{t+1}$
      Store transition $(s_t, \boldsymbol{\omega}_t, \tilde{a}_t, r_t, \mathbf{s}_{t+1})$ in $B$
      **for** $i = 0...N$ **do**
         Sample batch from replay buffer $B$
         Update actor network:
         $\theta \leftarrow \mathbb{E}^{\pi_\alpha^{\mathrm{mix}}}[(1 - \alpha)\nabla_a \boldsymbol{\omega}_{rb}^{\mathsf{T}} \boldsymbol{Q}(s, a, \boldsymbol{\omega}_{rb}; \phi)\nabla_\theta \pi(s, \boldsymbol{\omega}_{rb}; \theta)]$
         Update critic network:
         $\phi \qquad \leftarrow \qquad (1 - \beta)\,\nabla_\phi \mathbb{E}^{\pi_\alpha^{\mathrm{mix}}}\left[\|\boldsymbol{y}_{\boldsymbol{\omega}_{rb}} - \boldsymbol{Q}(s, a, \boldsymbol{\omega}_{rb}; \phi)\|_2^2\right] \qquad +$
         $\beta \nabla_\phi \mathbb{E}^{\pi_\alpha^{\mathrm{mix}}}\left[\|\boldsymbol{\omega}_{rb}^{\mathsf{T}} \boldsymbol{y}_{\boldsymbol{\omega}_{rb}} - \boldsymbol{\omega}_{rb}^{\mathsf{T}} \boldsymbol{Q}(s, a, \boldsymbol{\omega}_{rb}; \phi)\|_2^2\right]$
      **end for**
      Sample batch from replay buffer $B$
      Update adversary network:
      $\bar{\theta} \leftarrow \mathbb{E}^{\pi_\alpha^{\mathrm{mix}}}[\alpha \nabla_{\bar{a}} \boldsymbol{\omega}_{rb}^{\mathsf{T}} \boldsymbol{Q}(s, \bar{a}, \boldsymbol{\omega}_{rb}; \phi)\nabla_{\bar{\theta}} \bar{\pi}(s, \boldsymbol{\omega}_{rb}; \bar{\theta})]$
      Update the target networks:
         $\theta^- \leftarrow \tau\theta + (1 - \tau)\theta^-$
         $\bar{\theta}^- \leftarrow \tau\bar{\theta} + (1 - \tau)\bar{\theta}^-$
         $\phi^- \leftarrow \tau\phi + (1 - \tau)\phi^-$
   **end for**
**end for**

---

---

**Algorithm 4** multi-objective DDPG with linear scalarization function

---

**Input:** weighting coefficients $\beta$ and $\tau$
Randomly initialize actor $\pi(s, \boldsymbol{\omega}; \theta)$, adversary $\bar{\pi}(s, \boldsymbol{\omega}; \bar{\theta})$ and critic network $Q(s, a, \boldsymbol{\omega}; \phi)$
Initialize target networks with weights $\theta^-, \bar{\theta}^-$ and $\phi^-$
Initialize replay buffer $B$ and comprehensive metric $I$
**for** $episode = 0...M$ **do**
    Receive initial state $s_0$
    **for** $t = 0...T$ **do**
        Sample preference $\boldsymbol{\omega}_{ud}$ based on uniform-distribution:
        $\boldsymbol{\omega}_t \leftarrow \boldsymbol{\omega}_{ud}$
        Sample action $a_t = \pi(s_t, \boldsymbol{\omega}_t; \theta)$
        $\tilde{a}_t = a_t$ + exploration noise
        Execute action $\tilde{a}_t$ and observe reward $r_t$ and new state $s_{t+1}$
        Store transition $(s_t, \boldsymbol{\omega}_t, \tilde{a}_t, r_t, \mathbf{s}_{t+1})$ in $B$
        **for** $i = 0...N$ **do**
            Sample batch from replay buffer $B$
            Update actor network:
            $\theta \leftarrow \mathbb{E}^\pi[\nabla_a \boldsymbol{\omega}_{rb}^\mathsf{T} \boldsymbol{Q}(s, a, \boldsymbol{\omega}_{rb}; \phi) \nabla_\theta \pi(s, \boldsymbol{\omega}_{rb}; \theta)]$
            Update critic network:
$$\phi \leftarrow (1-\beta) \nabla_\phi \mathbb{E}^\pi \left[ \|\boldsymbol{y}_{\boldsymbol{\omega}_{rb}} - \boldsymbol{Q}(s, a, \boldsymbol{\omega}_{rb}; \phi)\|_2^2 \right] +$$
$$\beta \nabla_\phi \mathbb{E}^\pi \left[ \|\boldsymbol{\omega}_{rb}^\mathsf{T} \boldsymbol{y}_{\boldsymbol{\omega}_{rb}} - \boldsymbol{\omega}_{rb}^\mathsf{T} \boldsymbol{Q}(s, a, \boldsymbol{\omega}_{rb}; \phi)\|_2^2 \right]$$
        **end for**
        Update the target networks:
$$\theta^- \leftarrow \tau\theta + (1-\tau)\theta^-$$
$$\bar{\theta}^- \leftarrow \tau\bar{\theta} + (1-\tau)\bar{\theta}^-$$
$$\phi^- \leftarrow \tau\phi + (1-\tau)\phi^-$$
    **end for**
**end for**

---

### A.1.2 THEORETICAL DERIVATION

In this part, we provide the derivation details of some formulas.

The policy gradient of the protagonist based on $\boldsymbol{\omega}_{rb}$ can be derived:

$$
\begin{aligned}
\nabla_\theta L_{\boldsymbol{\omega}_{rb}}^\pi &\approx \mathbb{E}^{\pi_\alpha^{\mathrm{mix}}}[\nabla_\theta U_{\boldsymbol{\Omega}}(s, a^{\mathrm{mix}}, \boldsymbol{\omega}_{rb})] \\
&= \mathbb{E}^{\pi_\alpha^{\mathrm{mix}}}[\nabla_\theta \boldsymbol{\omega}_{rb}^\mathsf{T} \boldsymbol{Q}^{\pi_\alpha^{\mathrm{mix}}(\pi, \bar{\pi})}(s, a^{\mathrm{mix}}, \boldsymbol{\omega}_{rb}; \phi) + k \nabla_\theta M(s, a^{\mathrm{mix}}, \boldsymbol{\omega}_{rb})] \\
&= \mathbb{E}^{\pi_\alpha^{\mathrm{mix}}}[(1-\alpha)\nabla_a \boldsymbol{\omega}_{rb}^\mathsf{T} \boldsymbol{Q}(s, a, \boldsymbol{\omega}_{rb}; \phi) \nabla_\theta \pi(s, \boldsymbol{\omega}_{rb}; \theta) + k \nabla_a M(s, a, \boldsymbol{\omega}_{rb}) \nabla_\theta \pi(s, \boldsymbol{\omega}_{rb}; \theta)],
\end{aligned}
\tag{15}
$$

The policy gradient of the protagonist based on $\boldsymbol{\omega}_{bo}$ can be written as:

$$
\begin{aligned}
\nabla_\theta L_{\boldsymbol{\omega}_{bo}}^\pi &\approx \mathbb{E}^{\pi_\alpha^{\mathrm{mix}}}[\nabla_\theta U_{\boldsymbol{\Omega}}(s, a^{\mathrm{mix}}, \boldsymbol{\omega}_{bo})] \\
&= \mathbb{E}^{\pi_\alpha^{\mathrm{mix}}}[\nabla_\theta \boldsymbol{\omega}_{bo}^\mathsf{T} \boldsymbol{Q}^{\pi_\alpha^{\mathrm{mix}}(\pi, \bar{\pi})}(s, a^{\mathrm{mix}}, \boldsymbol{\omega}_{bo}; \phi) + k \nabla_\theta M(s, a^{\mathrm{mix}}, \boldsymbol{\omega}_{bo})] \\
&= \mathbb{E}^{\pi_\alpha^{\mathrm{mix}}}[(1-\alpha)\nabla_a \boldsymbol{\omega}_{bo}^\mathsf{T} \boldsymbol{Q}(s, a, \boldsymbol{\omega}_{bo}; \phi) \nabla_\theta \pi(s, \boldsymbol{\omega}_{bo}; \theta) + k \nabla_a M(s, a, \boldsymbol{\omega}_{bo}) \nabla_\theta \pi(s, \boldsymbol{\omega}_{bo}; \theta)],
\end{aligned}
\tag{16}
$$

The policy gradient of the adversary based on $\boldsymbol{\omega}_{rb}$ can be derived:

$$
\begin{aligned}
\nabla_{\bar{\theta}} L_{\boldsymbol{\omega}_{rb}}^{\bar{\pi}} &\approx \mathbb{E}^{\pi_\alpha^{\mathrm{mix}}}[\nabla_{\bar{\theta}} U_{\boldsymbol{\Omega}}(s, a^{\mathrm{mix}}, \boldsymbol{\omega}_{rb})] \\
&= \mathbb{E}^{\pi_\alpha^{\mathrm{mix}}}[\nabla_{\bar{\theta}} \boldsymbol{\omega}_{rb}^\mathsf{T} \boldsymbol{Q}^{\pi_\alpha^{\mathrm{mix}}(\pi, \bar{\pi})}(s, a^{\mathrm{mix}}, \boldsymbol{\omega}_{rb}; \phi) + k \nabla_{\bar{\theta}} M(s, a^{\mathrm{mix}}, \boldsymbol{\omega}_{rb})] \\
&= \mathbb{E}^{\pi_\alpha^{\mathrm{mix}}}[\alpha \nabla_{\bar{a}} \boldsymbol{\omega}_{rb}^\mathsf{T} \boldsymbol{Q}(s, \bar{a}, \boldsymbol{\omega}_{rb}; \phi) \nabla_{\bar{\theta}} \bar{\pi}(s, \boldsymbol{\omega}_{rb}; \bar{\theta}) + k \nabla_{\bar{a}} M(s, \bar{a}, \boldsymbol{\omega}_{rb}) \nabla_{\bar{\theta}} \bar{\pi}(s, \boldsymbol{\omega}_{rb}; \bar{\theta})],
\end{aligned}
\tag{17}
$$

The policy gradient of the adversary based on $\boldsymbol{\omega}_{bo}$ can be described as:

$$
\begin{aligned}
\nabla_{\bar{\theta}} L^{\bar{\pi}}_{\boldsymbol{\omega}_{bo}} &\approx \mathbb{E}^{\pi^{\mathrm{mix}}_\alpha}[\nabla_{\bar{\theta}} U_{\boldsymbol{\Omega}}(s, a^{\mathrm{mix}}, \boldsymbol{\omega}_{bo})] \\
&= \mathbb{E}^{\pi^{\mathrm{mix}}_\alpha}[\nabla_{\bar{\theta}} \boldsymbol{\omega}^{\mathsf{T}}_{bo} \boldsymbol{Q}^{\pi^{\mathrm{mix}}_\alpha (\pi,\bar{\pi})}(s, a^{\mathrm{mix}}, \boldsymbol{\omega}_{bo}; \phi) + k\nabla_{\bar{\theta}} M(s, a^{\mathrm{mix}}, \boldsymbol{\omega}_{bo})] \\
&= \mathbb{E}^{\pi^{\mathrm{mix}}_\alpha}[\alpha\nabla_{\bar{a}} \boldsymbol{\omega}^{\mathsf{T}}_{bo} \boldsymbol{Q}(s, \bar{a}, \boldsymbol{\omega}_{bo}; \phi)\nabla_{\bar{\theta}} \bar{\pi}(s, \boldsymbol{\omega}_{bo}; \bar{\theta}) + k\nabla_{\bar{a}} M(s, \bar{a}, \boldsymbol{\omega}_{bo})\nabla_{\bar{\theta}} \bar{\pi}(s, \boldsymbol{\omega}_{bo}; \bar{\theta})],
\end{aligned}
\tag{18}
$$

## A.2 DOMAIN

In this section, we give more details about training environment. We demonstrate our approach in four challenging continuous control domains 9.

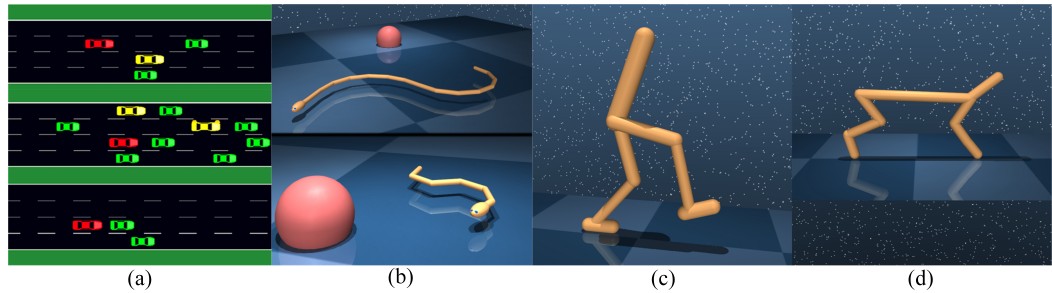

| (a) | (b) | (c) | (d) |

Figure 9: (a) Velocity control for autonomous electric vehicle in SUMO; (b) Swimmer environment in Mujoco; (c) Walker environment in Mujoco; (d) HalfCheetah environment in Mujoco.

### A.2.1 SUMO

SUMO is a free and open source traffic simulation suite, which can provide intelligent driving function verification environment. Transport efficiency and energy consumption are two conflicting objectives for autonomous electric vehicle. In this paper, a robust multi-objective longitudinal decision-making problem is focused to verify the effectiveness and advancedness of our proposed algorithm.

Observation and action space dimension: $\mathcal{S} \in \mathbb{R}^{16}, \mathcal{A} \in \mathbb{R}^1$.
The first objective is vehicle velocity:

$$
R_1 = \frac{v_x}{v_{max}},
$$

The second objective is energy conservation:

$$
R_2 = p\sqrt{1 - (\frac{v_x}{v_{max}})^2},
$$

where $v_x$ is vehicle longitudinal velocity, $v_{max}$ is maximum speed, $p = 1$ is scaling factor.

### A.2.2 SWIMMER-V2

Observation and action space dimension: $\mathcal{S} \in \mathbb{R}^8, \mathcal{A} \in \mathbb{R}^2$.

The first objective is running velocity:

$$
R_1 = |v_x|,
$$

The second objective is energy conservation:

$$
R_2 = \max(\sqrt{1 - \min(v_x, 1)^2} - 0.15\sum_i a_i^2, 0),
$$

where $v_x$ is longitudinal velocity, $a_i$ is the action of each actuator.

### A.2.3  WALKER2D-V2

Observation and action space dimension: $\mathcal{S} \in \mathbb{R}^{17}, \mathcal{A} \in \mathbb{R}^{6}$.

The first objective is running velocity:

$$R_1 = v_x + 1,$$

The second objective is energy conservation:

$$R_2 = 5 - \sum_i a_i^2,$$

where $v_x$ is longitudinal velocity, $a_i$ is the action of each actuator.

### A.2.4  HALFCHEETAH-V2

Observation and action space dimension: $\mathcal{S} \in \mathbb{R}^{17}, \mathcal{A} \in \mathbb{R}^{6}$.

The first objective is running velocity:

$$R_1 = \min(v_x, 4) + 1,$$

The second objective is energy conservation:

$$R_2 = 5 - \sum_i a_i^2,$$

where $v_x$ is longitudinal velocity, $a_i$ is the action of each actuator.

### A.3  IMPLEMENTATION

We implement the actor, adversary and critic neural networks by 2 fully connected hidden layers, which layer sizes are $\{256, 256\}$ and $\{512, 256\}$ in SUMO and Mujoco respectively.

Our scheme contains three important hyperparameters $\alpha$, $\lambda$ and $k$. $\alpha$ mainly affects the robustness of the policy model. If $\alpha$ is set to a smaller value, the robustness of the model will be reduced, otherwise, the performance of the model will be reduced. $\lambda$ and $k$ are responsible for balancing hypervolume, diversity and evenness metrics. if $\lambda$ and $k$ are set to larger values, then the diversity and evenness of the learned policies will be better, otherwise, the hypervolume will converge to a larger value. In order to select appropriate values, these three parameters need to be adjusted in the experiment.

The number of solution intervals $S_n$ is set to 9, and the main parameters of our BRMO-DDPG algorithm are reported in the Table 6.

Table 6: BRMO-DDPG Parameters.

| Parameter Name | Value |
|---|---|
| $\alpha$ | 0.1 |
| $\beta$ | 0.2 |
| $\gamma$ | 0.99 |
| $\lambda$ | 1 |
| $\tau$ | 0.01 |
| $k$ | 100 |
| $S_n$ | 9 |
| batch size | 128 |
| replay size | $1e^7$ |
| noise scale | 0.1 |
| learning rate for actor | $1e^{-4}$ |
| learning rate for critic | $1e^{-3}$ |
| learning rate for adversary | $1e^{-4}$ |

In the robustness and comparing the PGMORL tests, the utility is designed as follows:

$$u = \boldsymbol{\omega}^\intercal \boldsymbol{r} - \left\| \frac{\boldsymbol{r}}{\|\boldsymbol{r}\|_2} - \frac{\boldsymbol{\omega}}{\|\boldsymbol{\omega}\|_2} \right\|_2^2$$

We test the utility of the policy ten times under each mass and disturbance changes, and calculate the cumulative value of the utility. The mean and standard deviation of each policy's utility obtained from the whole test process are used to evaluate the performance of the policy under the environment uncertainty. The smaller standard deviation is, the stronger the robustness of the policy is. Moreover, for a policy , the higher the mean is, the stronger the capability to obtain utility under the environment uncertainty is. Hence, the robustness and utility in Table 5 and 7 refer to the standard deviation and the mean here.

In the experiment comparing the PGMORL scheme, we evenly selected ten trained policy models. Because we use one model for testing, we evenly selected ten preferences to input to the tested model. For the PGMORL method, we tested each model one hundred times and calculated the cumulative value of the utility. For BRMORL scheme, we tested each preference one hundred times, and then calculated the cumulative value of the utility. Therefore, the hypervolume and evenness in Table 5 and 7 are calculated by using the returns.

## A.4   Results

In this section, we give more experimental results.

### A.4.1   Swimmer-v2

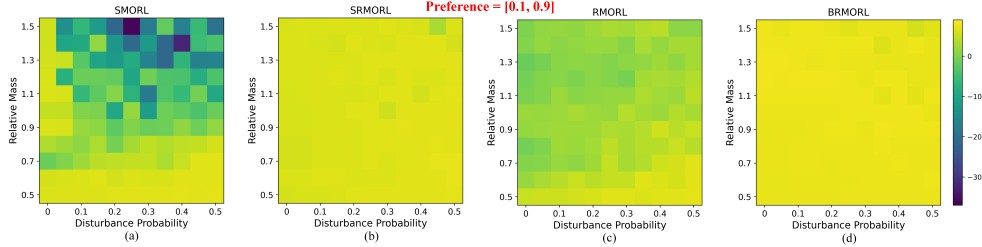

Figure 10: Robustness to environmental uncertainty on Swimmer-v2 domain. Disturbance probability represents the probability of a random disturbance being played instead of the selected action. Relative mass denotes the ratio of the current agent's mass to its original mass.

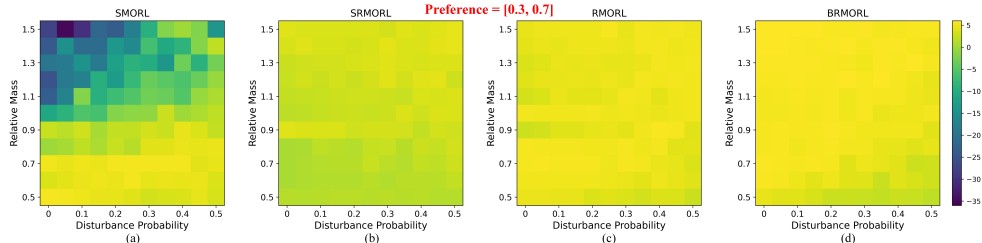

Figure 11: Robustness to environmental uncertainty on Swimmer-v2 domain.

### A.4.2   HalfCheetah-v2 and Walker2d-v2

The comparison results of the baseline (PGMORL) are provided here.



Figure 12: Robustness to environmental uncertainty on Swimmer-v2 domain.



Figure 13: Robustness to environmental uncertainty on Swimmer-v2 domain.

Table 7: Test results on HalfCheetah-v2.

|  | PGMORL | BRMORL |
|---|---|---|
| Hypervolume | 89984.28 | 21620.47 |
| Evenness | 0.31 | 0.36 |
| Robustness | 30.80 | 20.84 |
| Utility | -198.23 | -22.12 |

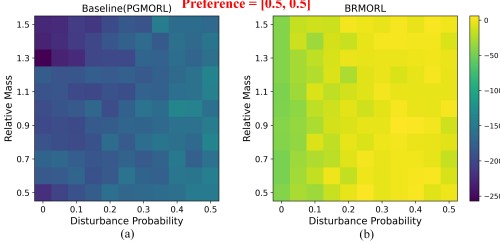

Figure 14: Robustness to environmental uncertainty on Walker2d-v2 domain. Disturbance probability represents the probability of a random disturbance being played instead of the selected action. Relative mass denotes the ratio of the current agent's mass to its original mass.

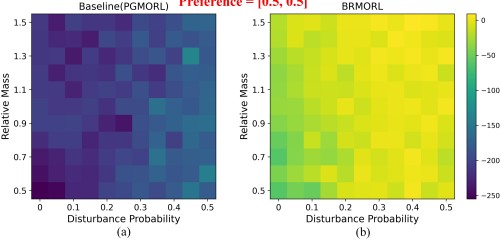

Figure 15: Robustness to environmental uncertainty on HalfCheetah-v2 domain.

