# OpenReview forum: "Approximating Pareto Frontier through Bayesian-optimization-directed Robust Multi-objective Reinforcement Learning"
_ICLR.cc/2021/Conference — Reject_

### Official Review · AnonReviewer2 · 2020-10-27
**Review of Approximating Pareto Frontier through Bayesian-optimization-directed Robust Multi-objective Reinforcement Learning**

**Rating:** 5
**Confidence:** 4

**Review:**

This paper proposes a framework to tackle uncertainty in multi-objective optimization of reinforcement learning problems. Uncertainty is represented as an adversary over preferences. Fitness is measured by a multi-objective quality indicators while Bayesian optimization is used to bring improvements. The proposed method is evaluated on four benchmark problems.

While the page limit makes it difficult to detail all the different aspect of the contribution, the clarity still needs to be improved. For instance, it is worth adding a detailed algorithmic description of the approach. At various stages there are many different options, so the specific choices should be better introduced and limits discussed.

Detailed comments and questions:

a) There are many parameters to set (alpha, beta, omega, lambda), how sensitive are they in practice? They are not mentioned in the experimental part, hence the results cannot be reproduced.

b) Nash games are defined over partitions of the design variables, but it does not seem to be the case here?

c) Robustness can be defined in many different ways (expectation/variance trade-off, quantiles, worst case, chance constraints). How does it differ from optimizing the worst case here?

d) Pareto front quality indicators are widely studied in the multi-objective optimization literature, existing ones should be reviewed first. See e.g.,:
- Charles Audet, J Bigeon, D Cartier, Sébastien Le Digabel, Ludovic Salomon. Performance indicators in multiobjective optimization. 2020. ⟨hal-02464750⟩
- Cheng S., Shi Y., Qin Q. (2012) On the Performance Metrics of Multiobjective Optimization. In: Tan Y., Shi Y., Ji Z. (eds) Advances in Swarm Intelligence. ICSI 2012. Lecture Notes in Computer Science, vol 7331. Springer, Berlin, Heidelberg. https://doi.org/10.1007/978-3-642-30976-2_61

e) Pareto based acquisition functions are generally preferred in Bayesian MOO compared to scalar ones, such as the one proposed here on a GP fit of the defined utility. See for instance:
- Emmerich, M. T.; Deutz, A. H. & Klinkenberg, J. W. Hypervolume-based expected improvement: Monotonicity properties and exact computation. Evolutionary Computation (CEC), 2011 IEEE Congress on, 2011, 2147-2154
- Picheny, V. Multiobjective optimization using Gaussian process emulators via stepwise uncertainty reduction Statistics and Computing, Springer, 2015, 25, 1265-1280
- Predictive Entropy Search for Multi-objective Bayesian Optimization. Daniel Hernandez-Lobato,  Jose Hernandez-Lobato,  Amar Shah,  Ryan Adams ; PMLR 48:1492-1501.

Additional related references of interest from the literature that could be discussed:
- Lepird, J. R.; Owen, M. P. & Kochenderfer, M. J., Bayesian preference elicitation for multiobjective engineering design optimization, Journal of Aerospace Information Systems, American Institute of Aeronautics and Astronautics, 2015, 12, 634-645
- Paria, B.; Kandasamy, K. & Póczos, B. A flexible framework for multi-objective bayesian optimization using random scalarizations Proceedings of The 35th Uncertainty in Artificial Intelligence Conference, PMLR 115:766-776, 2020.
- Multi-attribute Bayesian optimization with interactive preference learning. R Astudillo, P Frazier
International Conference on Artificial Intelligence and Statistics, 4496-4507


f) Fig. 3: either a point is Pareto optimal, or it is not. All blue and violet solutions are dominated by the green ones, so the green Pareto front is better.

g) Fig. 1 lacks a detailed description. Are the different panel successive iterations for instance?

h) Only bi-objective examples are presented, how does it scale with more objectives?

Typos:
P1: algorithms have demonstrated its worth → their
Fig. 2: can not parallel

## Post rebuttal comments
The authors largely modified the paper according to the comments, with a lot of additional content. While this is quite beneficial, the paper raised many questions, some of which may need further treatment (for instance, increasing the number of objectives has an effect on the number of Pareto optimal solutions that is is not trivial).

---

> ### Author Response · Authors · 2020-11-23
> **Answer the experts' questions and explain the revision of the paper**
>
> 6. Reviewer: Fig. 3: either a point is Pareto optimal, or it is not. All blue and violet solutions are dominated by the green ones, so the green Pareto front is better.
>
> Author: Thank you for your valuable comments. We have made further improvements in the current version. In Section 4.4, We added a more accurate and detailed descriptions.
> In fact, what we want to illustrate is similar to the situation in Figure 6(c) (the current version). For the same problem, the Pareto front approximated by different algorithms is generally different, and the real Pareto front is also unknown. In Figure 4 (the current version), the Pareto frontiers 1, 2 and 3 are obtained by different algorithms, instead of same algorithm.
> From the perspective of the multi-objective optimization, Pareto front 1 is indeed better than the other two "Pareto fronts".
> In practice, however, we think that sometimes more effective options may be more important. For example, an electric car participating in a competition will pay more attention to speed rather than energy saving; energy conservation would be more important if the electric car had little residual power; in many cases, when driving this electric car, we have to make a tradeoff between speed and energy efficiency. Therefore, we hope that our trained model can approximate the "good"(robust, diverse, well-distributed, Pareto optimal or even suboptimal) policy for any specified preference.
>
> 7. Reviewer: Fig. 1 lacks a detailed description. Are the different panel successive iterations for instance?
>
> Author: Thank you for your valuable comments. We have made further improvements in the current version.  In Seciton 4.1, we have described Figure 2 (the current version) in more detail. The different panel are successive iterations in Figure 2 (the current version).
>
> 8. Reviewer: Only bi-objective examples are presented, how does it scale with more objectives?
>
> Author: Thank you for your valuable comments. It is very easy to solve the more objectives problem with our algorithm. We only need to set the preference vector to more dimensions, and there is no change in other places.
>
> 9. Reviewer: Typos: P1: algorithms have demonstrated its worth → their Fig. 2: can not parallel.
>
> Author: Thank you for your valuable comments. We have made further improvements in the current version. All of these issues have been addressed.

---

> ### Author Response · Authors · 2020-11-23
> **Answer the experts' questions and explain the revision of the paper**
>
> We appreciate very much for your time and valuable comments in reviewing our work. We have made a lot of improvements in the current version.
>
> 1. Reviewer:  a) There are many parameters to set (alpha, beta, omega, lambda), how sensitive are they in practice? They are not mentioned in the experimental part, hence the results cannot be reproduced.
>
> Author: In Appendix, the effect of important hyperparameters has been explained in detail, and the values are given in Table 6.
> α mainly affects the robustness of the policy model. If α is set to a smaller value, the robustness of the model will be reduced, otherwise, the performance of the model will be reduced.
> λ and k are responsible for balancing hypervolume, diversity and evenness metrics. if λ and k are set to larger values, then the diversity and evenness of the learned policies will be better, otherwise, the hypervolume will converge to a larger value
> Increasing the value of β makes the landscape of loss function smooth. For details, please see the Reference "A Generalized Algorithm for Multi-Objective Reinforcement Learning and Policy Adaptation".
> In order to select appropriate values, these three parameters need to be adjusted in the experiment.
>
> 2. Reviewer: Nash games are defined over partitions of the design variables, but it does not seem to be the case here?
>
> Author: Thank you for your valuable comments. In our paper, we use a special zero-sum game between a protagonist and an adversary. For details, please see the Reference "Action Robust Reinforcement Learning and Applications in Continuous
> Control".
>
> 3. Reviewer: Robustness can be defined in many different ways (expectation/variance trade-off, quantiles, worst case, chance constraints). How does it differ from optimizing the worst case here?
>
> Author: Thank you for your valuable comments. Compared with the robust setting based on the worst-case, our robust setting can adjust the opponent's attack strength or worst degree through the parameter α.
>
> 4. Reviewer: Pareto front quality indicators are widely studied in the multi-objective optimization literature, existing ones should be reviewed first.
>
> Author: Thank you for your valuable comments. We have made further improvements in the current version. In Section 4.4, the reference papers provided have been summarized and cited.
>
> 5. Reviewer: Pareto based acquisition functions are generally preferred in Bayesian MOO compared to scalar ones, such as the one proposed here on a GP fit of the defined utility； Additional related references of interest from the literature that could be discussed.
>
> Author: Thank you for your valuable comments. Our paper focuses on robust multi-objective reinforcement learning problem, and Bayesian optimization algorithm is adopted to guide the agent to evolve towards improving the quality of the approximated Pareto frontier. The six papers listed are all about multi-objective optimization, which is different from multi-objective reinforcement learning. The goal in reinforcement learning is to determine a policy which maximizes expected return when the agent acts according to it, which is sequential decision-making problem. Optimization problem can be defined as determining a solution that are within the feasible region to minimize (maximize) an objective function, which is a broader domain than reinforcement learning. Because this paper focuses on reinforcement learning problem and the paper has strict space limitation, we mainly introduce the researches of multi-objective reinforcement learning and robust reinforcement learning in Section 2 (Related work).
> Of course, if the listed literatures about Bayesian multi-objective optimization is introduced, our work will be better. But there is really no room for this paper, so we can only introduce the most directly related researches. Thank you for your valuable suggestions.

---

### Official Review · AnonReviewer4 · 2020-10-28
**Interesting idea but many details are missing**

**Rating:** 5
**Confidence:** 4

**Review:**

The paper proposes a robust multi-objective RL approach and a non-linear utility metric to enforce an accurate and evenly distributed representation of the Pareto frontier. Robustness is obtained by formulating the problem as a two-player zero-sum game. The goal of the main agent is thus to learn the policies on the Pareto frontier under attacks from the adversary. This is achieved by training a single network to generate approximate Pareto optimal policies for any provided preference. To train this network, they introduce a new metric for Pareto frontier evaluation based on hypervolume and entropy (to force evenly distributed solutions). The resulting algorithm has the classical structure of an actor-critic algorithm where the critic provides an estimate of the Q-function and the actor updates the policies of the protagonist and adversary through alternate optimization.

Could you give more motivations why a robust approach is needed for MORL? The motivation now seems simply to be that the literature didn't take into account robustness.

I think the idea of the paper is quite interesting but it is not well written/explained. I think a few details are missing.

For example, in section 4.3.1 it is not clear to me how the loss is constructed. You mentioned that the objective is to learn the Q-function of the protagonist but the target value $y$ is built using the mix policy and Q-function. Could you clarify the meaning of this loss?
There are also two different definitions of $y$ ($s'$ vs $s$ and $\omega'$ vs $\omega$).

What is the meaning of $\mathbb{E}^{\pi^{mix}}$? Does it mean expectation wrt to the stationary distribution induced by the policy?
It is important for understanding equations 8 and 9. Why there is an approximation in the definition of the gradients? Shouldn't be $\nabla \mathbb{E}^{\pi}[]$?

Concerning section 4.4, you compare with the metric proposed in (Xu et al. 2020) without explaining it. It is hard for the reader to understand why it is not a good metric without knowing the metric. In general, as you acknowledged, a good approximate Pareto frontier should be accurate, evenly distributed and have a covering similar to the one of the true Pareto frontier. These properties are not equally important. This is to say that I think the example in figure 3 has a major drawback. Frontier 2 and 3 are not Pareto frontier since are dominated by 3. Not sure that everyone will agree that 2 is better than 3. I suggest you change the example.

To overcome this, you introduced an entropic measure based on a partitioning of the Pareto frontier. There is no mention of how to do that in practice. Could you explain it? Are you partitioning the space of preferences?
A standard measure to enforce spread solutions is the crowding distance (eg in your reference Parisi et al. 2017 and many more), could you apply the same idea of interval partition on this?
Could you explain why you introduced evenness as a multiplicative factor rather than an additive one in $I(P)$?


Experiments. There is no information about the implementation and parameters/configurations used. It is thus very difficult to parse the results. For example, how are preferences selected for standard methods (ie all expect BRMORL)?
Why is the comparison with (Yang et al. 2019) missing? I think this is a relevant algorithm for the setting.
You should add an explicit reference to the papers introducing the methods in Table 3, ie RA, PFA, MOEA/D and META.

I think the paper contains an interesting idea but, given the mentioned concerns, I think this is a borderline paper. I'm looking forward to the authors' feedback.

Minor issues
-I think it's more precise to define \mathcal{A}^{mix} as the space of possible combinations of actions. The probability $\alpha$ should be associated with $\pi^{mix}$ rather than with the action space.
-Reference missing to SUMO
-You use both $\mathbb{E}_\pi$ and $\mathbb{E}^{\pi, \pi'}$ but these symbols are never explained
-Figure 2:  "OA can not parallel to OB"
-You use often the term convergence in the description of the algorithm. How do you evaluate convergence?
-In figure 4 what is $\alpha(\Omega)$? Is it related to $\alpha$ used in the definition of the mix policy?

---

> ### Author Response · Authors · 2020-11-23
> **Answer the experts' questions and explain the revision of the paper**
>
> 6. Reviewer: A standard measure to enforce spread solutions is the crowding distance (eg in your reference Parisi et al. 2017 and many more), could you apply the same idea of interval partition on this?
>
> Author: Thank you for your valuable comments. This is a very good suggestion and idea, but due to the time constraints for improving the paper, we will try to work on this indicator to improve our method in the future.
>
> 7. Reviewer: Could you explain why you introduced evenness as a multiplicative factor rather than an additive one in I(P)?
>
> Author: Thank you for your valuable comments. With the increase of training times, hypervolume may be a large value. In order to ensure the evenness (0~1) is effective, multiplication is used here.
>
> 8. Reviewer: Experiments. There is no information about the implementation and parameters/configurations used. It is thus very difficult to parse the results. For example, how are preferences selected for standard methods (ie all expect BRMORL)? Why is the comparison with (Yang et al. 2019) missing?
>
> Author: Thank you for your valuable comments. We have made further improvements in the current version. In Appendix A.3, the implementation and setup details of the algorithm and the experiment are supplemented. Moreover, in Appendix A.1.1, pseudocodes for brmorl, rmorl, srmorl and smorl has been added.
> The method of (Yang et al. 2019) is verified in the environment based on discrete action. It will be our future work to verify the performance of our scheme in the environment based on discrete action.
>
> 9. Reviewer: I think it's more precise to define \mathcal{A}^{mix} as the space of possible combinations of actions. The probability  should be associated with  rather than with the action space. -Reference missing to SUMO -You use both  and  but these symbols are never explained -Figure 2: "OA can not parallel to OB" -You use often the term convergence in the description of the algorithm. How do you evaluate convergence? -In figure 4 what is α(Ω)? Is it related to  used in the definition of the mix policy?
>
> Author: Thank you for your valuable comments. We have made further improvements in the current version. All of these issues have been addressed.
> In this paper, we use hypervolume to evaluate the convergence of the algorithm.
> To avoid misunderstanding, we have modified α(Ω) as U(Ω) in Figure 5 (the current version). U(Ω) is the utility of the candidate set (preference set in our task).

---

> > ### Comment · AnonReviewer4 · 2020-11-24
> > **Response**
> >
> > Thank you for the numerous comments. The current version has sensibly changed compared to the original version. It would have been better to highlight the changes in the new version (eg with a different color).
> >
> > I have two quick comments:
> > 1) *About the example in Figure 3.*
> > The distribution of solutions on frontier 1 is well distributed (accordingly to the image). Are you referring to the fact that the frontier is not evenly distributed accordingly to the weights of the linear scalarization? If it is the case, you should clarify it. Could you support with references the claim that frontier 2 is better than frontier 1 for practitioners?
> >
> > 2) *Comparison with state-of-the-art missing.*
> > In the new version, you removed the comparison with state-of-the-art. I think it is important and necessary to compare with existing algorithms.

---

> > > ### Author Response · Authors · 2020-11-24
> > > **Response to reviewer**
> > >
> > > We appreciate very much for your time and valuable comments in reviewing our work.
> > >
> > > 1. reviewer: Comparison with state-of-the-art missing. In the new version, you removed the comparison with state-of-the-art. I think it is important and necessary to compare with existing algorithms.
> > >
> > > Author: The comparison experiment with state-of-the-art algorithm is not missing, and there are further supplements. Please see Table 5, Table 7, Figure 14 and Figure 15.  More details and results are given in Experiment Section and Appendix Section.

---

> > > > ### Comment · AnonReviewer4 · 2020-11-24
> > > > **Response**
> > > >
> > > > Other questions:
> > > > - What about the other algorithms (RA, PFA, MOEA/D, RANDOM, META) that were reported in the original version?
> > > > - Why did you remove them?
> > > > - Why PGMORL is tested only on two domains?

---

> > > > > ### Author Response · Authors · 2020-11-24
> > > > > **Response to reviewer**
> > > > >
> > > > > 1. reviewer: What about the other algorithms (RA, PFA, MOEA/D, RANDOM, META) that were reported in the original version? Why did you remove them?
> > > > >
> > > > > Author: Thank you for your valuable comments. Table 3 (the old version) has been deleted. Because the work of Xu et al. (2020) has compared those baseline methods (RA, PFA, MOEA/D, RANDOM, META), we only need to compare the scheme of Xu et al. (2020) in this paper.
> > > > >
> > > > > 2. reviewer: Why PGMORL is tested only on two domains?
> > > > >
> > > > > Author: Thank you for your valuable comments. We think these two test environments should prove that our scheme is efficient and advanced. Of course, it would be better if we could verify the performance of our scheme in more test environments. However, the space and modification time of the paper are very limited, so we will verify the performance of our method in more environments in the future.
> > > > >
> > > > > Thanks again for your valuable comments.

---

> > > ### Author Response · Authors · 2020-11-24
> > > **Response to reviewer**
> > >
> > > We appreciate very much for your time and valuable comments in reviewing our work.
> > >
> > > 1. reviewer: About the example in Figure 3. The distribution of solutions on frontier 1 is well distributed (accordingly to the image). Are you referring to the fact that the frontier is not evenly distributed accordingly to the weights of the linear scalarization? If it is the case, you should clarify it. Could you support with references the claim that frontier 2 is better than frontier 1 for practitioners?
> > >
> > > Author: In Section 4.4, We added a more accurate and detailed descriptions. In fact, what we want to illustrate is similar to the situation in Figure 6(c) (the latest version). For the same problem, the Pareto front approximated by different algorithms is generally different, and the real Pareto front is also unknown. In Figure 4 (the latest version), the Pareto frontiers 1, 2 and 3 are obtained by different algorithms, instead of same algorithm. From the perspective of the multi-objective optimization, Pareto front 1 is indeed better than the other two "Pareto fronts". In practice, however, we think that sometimes more effective options may be more important. For example, an electric car participating in a competition will pay more attention to speed rather than energy saving; energy conservation would be more important if the electric car had little residual power; in many cases, when driving this electric car, we have to make a tradeoff between speed and energy efficiency. Therefore, we hope that our trained model can approximate the "good"(robust, diverse, well-distributed, Pareto optimal or even suboptimal) policy for any specified preference.
> > >
> > > In Figure 4 (the latest version),  the main problem reflected in Figure 1 is the lack of diversity of solutions on frontier 1.
> > >
> > > According to Proposition 1 and Proposition 2 in Section 4.4, the  evenness metric we propose can simultaneously evaluate the diversity and uniformity of the solutions on Pareto frontier, and the experiments also verified that.
> > >
> > > The following references highlight the importance of policy diversity for MORL. See e.g.,:
> > > "A Generalized Algorithm for Multi-Objective Reinforcement Learning and Policy Adaptation",
> > > “Meta-Learning for Multi-objective Reinforcement Learning”.

---

> > > ### Author Response · Authors · 2020-11-24
> > > **Response to reviewer**
> > >
> > > Thank you for your valuable comments.
> > >
> > > We have made a lot of improvements in the latest version. We added a large number of the details for algorithms and experiments, added more detailed supplementary tests, and optimized models, and so on.
> > >
> > > Moreover, we have studied the reviewers’ comments carefully, and a thorough revision has been made accordingly (marked in yellow in the paper).

---

> > > ### Author Response · Authors · 2020-11-25
> > > **Response to reviewer**
> > >
> > > Our view is to satisfy that the policy is robust, diverse and well-distributed, sometimes it is acceptable that the policy is not Pareto optimal. Although our approach cannot guarantee the learned policy is optimal, it is approximately robust Pareto optimal. This is like classical robust control, in order to ensure that the control system is robust to perturbation, control system need to sacrifice part of the performance to obtain the robust optimal solution.

---

> ### Author Response · Authors · 2020-11-23
> **Answer the experts' questions and explain the revision of the paper**
>
> We appreciate very much for your time and valuable comments in reviewing our work. We have made a lot of improvements in the current version.
>
> 1. Reviewer: Could you give more motivations why a robust approach is needed for MORL? The motivation now seems simply to be that the literature didn't take into account robustness.
>
> Author: In the introduction of the latest edition of the paper, we have added Figure 1 and described the motivation our work in more detail. "Preference and uncertainty jointly affect the decision-making behavior of the agent."
>
> 2. Reviewer: In section 4.3.1 it is not clear to me how the loss is constructed.  Could you clarify the meaning of this loss?
>
> Author: Thank you for your valuable comments. In the current version, we have made further improvements. In Section 4.3.1 and Appendix A.1.1, more detailed descriptions and derivations are given.
> Equation 7 is a target vectorized Q-function considering adversary attack. The loss_1 (Equation 8) is to train the critic network to approximate the vectorized value function considering uncertainty. The auxiliary loss setting (Equation 9) is to make the landscape of loss function smooth, please see the Reference "A Generalized Algorithm for Multi-Objective Reinforcement Learning and Policy Adaptation" for details.
>
> 3. Reviewer: What is the meaning of E^π^mix_α? Does it mean expectation wrt to the stationary distribution induced by the policy? It is important for understanding equations 8 and 9. Why there is an approximation in the definition of the gradients? Shouldn't be?
>
> Author: Thank you for your valuable comments. E^π^mix_α means the calculating expectation under the mixed policy based on a protagonist and an adversary.
> Because the policy gradient is estimated based on sampled data, there is an approximation in the definition of the gradients. In Section 4.3, more derivations and descriptions have been added.
>
> 4. Reviewer: Concerning section 4.4, you compare with the metric proposed in (Xu et al. 2020) without explaining it. It is hard for the reader to understand why it is not a good metric without knowing the metric.
>
> Author: Thank you for your valuable comments. In the current version, we have made further improvements. In Section 4.4, more detailed descriptions are added.
> The metric proposed in (Xu et al. 2020) is a comprehensive index considering  hypervolume metric and sparsity metric.
>
> 5. Reviewer: To overcome this, you introduced an entropic measure based on a partitioning of the Pareto frontier. There is no mention of how to do that in practice. Could you explain it? Are you partitioning the space of preferences?
>
> Author: Thank you for your valuable comments. We have made further improvements in the current version. In Section 4.4, Sn is the number of solution intervals, the value of Sn is set to 9, which has been given in Table 6 of Appendix. We divide the preference space into 9 equal intervals based on Sn.

---

### Official Review · AnonReviewer1 · 2020-10-28
**Interesting problem, but lacking clarity and motivation**

**Rating:** 5
**Confidence:** 4

**Review:**

Summary:
This paper seeks to train multi-objective RL policies that are robust to environmental uncertainties. There are two main contributions: a novel approach to solve this problem, and a novel metric to evaluate Pareto fronts. The metric combines the typical hypervolume metric (that captures the quality/performance of a Pareto front) with a novel "evenness" metric, that captures how well solutions are spread out across the space of preferences. The proposed approach, called BRMORL, consists of training a protagonist policy that maximizes utility alongside an adversarial policy that seeks to minimize utility (motivated by zero-sum game theory), while using Bayesian optimization to select preferences to train on, in order to optimize the hypervolume-and-evennesss metric. Both the protagonist and adversarial policy are conditioned on preferences.

Recommendation:
This paper connects two seemingly orthogonal problems, multi-objective RL and robustness. This is an interesting topic, but there are several issues regarding clarity and the motivation (as detailed in the cons list below). I think this paper could be a valuable contribution for MORL, but _not_ for MORL that is robust to environmental uncertainty, which is what the claim is. Thus I recommend rejection.

Pros:
* Training policies that are robust _and_ flexibly trade off between preferences is an interesting and relevant problem.
* The empirical evaluation shows that the approach outperforms ablations and an existing state-of-the-art MORL approach (Xu et al. 2020) on continuous control tasks.

Cons:
* Clarity: the introduction should clearly define what _robustness_ means. Currently it's unclear what problem this paper is trying to solve. Does the approach try to achieve robustness to environment dynamics / perturbations, or robustness across preferences, or both? My interpretation is that robustness refers to both kinds. I can understand how BRMORL would improve robustness across preferences, and perhaps also perturbations, but am skeptical about whether it improves robustness to environment dynamics (see next point).
* The motivation behind this approach is questionable: I'm not convinced that BRMORL actually leads to training policies that are more robust, with respect to environment dynamics or perturbations. This is not shown clearly in the empirical evaluation, and also is not obvious from the approach itself. I don't see the connection between having an adversarial policy and being robust to the dynamics of the system (e.g., masses of limbs). Figure 6 shows that BRMORL has better robustness to environmental uncertainty than SMORL, but that could just be because SMORL is the worst-performing ablation, and just doesn't find particularly high-performing policies (as shown in Figure 5c). How does BRMORL compare to RMORL or SRMORL?
* It would help to have an algorithm box for BRMORL, that clarifies how the adversary policy, protagonist policy, and Bayesian optimization are used to gather data and for training.
* The proposed metric is questionable. The goal is to capture both diversity and quality of solutions, but in Figure 3, I would argue that Pareto front 1 is indeed better, because these points dominate _all_ of the points on Pareto fronts 2 and 3, and the purpose of MORL is to find non-dominated policies.
* The chosen scalar utility function $U$ is not properly justified. In particular, does $M$ (in Equation 2) still make sense when the objectives have significantly different reward scales (e.g., if one objective's return is typically from 0 to 10, and the other's is from 10 to 100)? Even after normalizing, the Q-value term will only be in a portion of the first quadrant, whereas the $w$ term can cover the entire first quadrant.
* Unjustified hyperparameters for trading off between terms in the losses: $k$ in the scalar utility function, $\beta$ for the two terms in the Q-function loss, and $\lambda$ for the comprehensive metric that combines hypervolume and evenness. How should these be chosen?
* The Related Work doesn't give enough credit to existing MORL approaches. First, Xu et al. (2020) is actually able to find a well-distributed set of Pareto-optimal solutions. In addition, existing methods are stated to only be able to find solutions on the convex portions of a Pareto front. Bringing up this point implies that BRMORL does better (i.e., is able to find solutions on concave portions of the Pareto front), but this is not shown empirically. Finally, the related work states that most existing approaches are only applied to domains with discrete action spaces. It should acknowledge that both Abdolmaleki et al. (2020) and Xu et al. (2020) are applied to high-dimensional continuous control tasks.
* Lack of experimental details for reproducibility, e.g., network architetures and DDPG hyperparameters.

Other comments:
* There are quite a few grammatical errors and typos throughout the paper.
* Definition 3 is imprecise. First, is $a$ a policy or an action? It seems like it should be a policy because it's a member of the policy set, but it's used to denote actions in the previous section, Section 3.1. Also, why are $I$ and $II$ included in the game definition, when they are already represented by the policy sets?
* There is not enough explanation given for Figure 1. Where do the uniformly-sampled preferences come from (the gray dashed lines)? What is the "optimal guess point"? Does Bayesian optimization only suggest one preference at a time (in red)? What is the acquisition function? (This is defined too late in the paper, and only in the caption for Figure 4.)
* It would be more accurate to make the $k$ explicit in equations 8 and 9, because it's different in $M(\cdot)$ for the two equations, but the current notation implies it's the same.
* In the empirical evaluation, SRMORL, an ablation of BRMORL, finds policies that dominate those found by BRMORL (Figure 5c). How can this be interpreted / explained?
* Table 3 needs an accompanying explanation of the different MORL methods.

---

> ### Author Response · Authors · 2020-11-23
> **Answer the experts' questions and explain the revision of the paper**
>
> 8. Reviewer: Bringing up this point implies that BRMORL does better (i.e., is able to find solutions on concave portions of the Pareto front).
>
> Author: Thank you for your valuable comments. In the current version, we have made further improvements. It can also be found from Figure 7(c) the BRMORL mothod is not only able to find solutions on the convex portions of the Pareto frontier, but also the concave portions.
>
> 9. Reviewer: Lack of experimental details for reproducibility, e.g., network architetures and DDPG hyperparameters.
>
> Author: Thank you for your valuable comments. In the current version, we have made further improvements. A large number of algorithms and experimental details are supplemented in Appendix.
>
> 10. Reviewer: There are quite a few grammatical errors and typos throughout the paper. Definition 3 is imprecise.
>
> Author: Thank you for your valuable comments. In the current version, we have made further improvements, and the Definition 3 (the old version) has been deleted.
>
> 11. Reviewer: There is not enough explanation given for Figure 1. Where do the uniformly-sampled preferences come from (the gray dashed lines)? What is the "optimal guess point"? Does Bayesian optimization only suggest one preference at a time (in red)? What is the acquisition function? (This is defined too late in the paper, and only in the caption for Figure 4.).
>
> Author: Thank you for your valuable comments. We have made further improvements in the current version. In Seciton 4.1, we have described Figure 2 (the current version) in more detail.
> In our scheme, in the data collection stage, part of the preference comes from the BO algorithm and part of the preference comes from the uniform distribution; in the model training phase, part of the preference comes from the BO algorithm, and part of the preference comes from the replay buffer.  For more details, see Algorithm 1 in Appendix.
> Using the Bayesian model, acquisition function (Frazier, 2018) can determine optimal guess point, which is the suggested preference in our task.
>
> 12. Reviewer: It would be more accurate to make the explicit in equations 8 and 9, because it's different in for the two equations, but the current notation implies it's the same.
>
> Author: Thank you for your valuable comments. We have made further improvements in the current version. For more details, see Equations 3 and 4 in Section 4.2.
>
> 13. Reviewer: In the empirical evaluation, SRMORL, an ablation of BRMORL, finds policies that dominate those found by BRMORL (Figure 5c). How can this be interpreted / explained?
>
> Author: Thank you for your valuable comments. In Figure 6(c) (the current version), SRMORL allocates more computing resources to optimize a small number of preference, so that the policies in that part of preference interval is better, but it also causes the overfitting of the model in this part of the preference interval, which leads to no Pareto optimal policy in other preference intervals.
>
> 14. Reviewer: Table 3 needs an accompanying explanation of the different MORL methods.
>
> Author: Thank you for your valuable comments. In the current version, we have made further improvements. Table 3 (the old version) has been deleted. Because the work of Xu et al. (2020) has compared those baseline methods, we only need to compare the scheme of Xu et al. (2020) in this paper.

---

> ### Author Response · Authors · 2020-11-23
> **Answer the experts' questions and explain the revision of the paper**
>
> 5. Reviewer: Does (in Equation 2) still make sense when the objectives have significantly different reward scales? Even after normalizing, the Q-value term will only be in a portion of the first quadrant, whereas the w term can cover the entire first quadrant.
>
> Author: Thank you for your valuable comments. It still makes sense. A vector divided by its own 2-norm is to convert this vector into a unit vector. Therefore, the objectives of different scales have no effect on Equation 4 (the current version).
> The purpose of Equation 4 (the current version) is to make the Q-value vector parallel to the preference vector. The core idea here is that the difference between two parallel unit vectors is a zero vector.
>
>
> 6. Reviewer: Unjustified hyperparameters for trading off between terms in the losses, How should these be chosen?
>
> Author: Thank you for your valuable comments. In the current version, we have made further improvements. In Appendix A.3, the effect of important hyperparameters has been explained in detail, and the values are given in Table 6.
> α mainly affects the robustness of the policy model. If α is set to a smaller value, the robustness of the model will be reduced, otherwise, the performance of the model will be reduced.
> λ and k are responsible for balancing hypervolume, diversity and evenness metrics. if λ and k are set to larger values, then the diversity and evenness of the learned policies will be better, otherwise, the hypervolume will converge to a larger value
> Increasing the value of β makes the landscape of loss function smooth. For details, see the Reference "A Generalized Algorithm for Multi-Objective Reinforcement Learning and Policy Adaptation".
> In order to select appropriate values, these three parameters need to be adjusted in the experiment.
>
> 7. Reviewer: Xu et al. (2020) is actually able to find a well-distributed set of Pareto-optimal solutions.
>
> Author: Thank you for your valuable comments. Although the work of Xu et al. (2020) is able to find a well-distributed set of Pareto-optimal solutions, their work requires multiple policy models to generate good representation. Our algorithm only needs a policy model to approximate a good representation for Pareto frontier. Moreover, in Table 5, we compared our BRMORL scheme with state-of-the-art baseline (PGMORL) provided by Xu et al. (2020). Although our method is not superior in hypervolume, it outperforms PGMORL method in evenness, robustness and utility. More results can be found in Appendix A.4.2.

---

> ### Author Response · Authors · 2020-11-23
> **Answer the experts' questions and explain the revision of the paper**
>
> We appreciate very much for your time and valuable comments in reviewing our work. We have made a lot of improvements in the current version.
>
> 1. Reviewer: The introduction should clearly define what robustness means. Currently it's unclear what problem this paper is trying to solve. Does the approach try to achieve robustness to environment dynamics/perturbations, or robustness across preferences, or both? The motivation behind this approach is questionable, I'm not convinced that BRMORL actually leads to training policies that are more robust.
>
> Author: In the introduction of the latest edition of the paper, we have clearly explained the robustness of policy. "a policy is said to be robust if its capability to obtain utility is relatively stable under environmental changes."
> In the experiment section of the latest edition of the paper, the standard deviation based on the utility of the policy is adopted to quantify the robustness. The stronger the robustness of a policy is, then the smaller its standard deviation is.
> We test with jointly varying both mass (environment dynamics/perturbations) and disturbance probability.
> The results of quantitative analysis and visualization of robustness across preferences show the effectiveness of our method in Section 5 and Appendix A.4.
> In the introduction of the latest edition of the paper, we have added Figure 1 and described the motivation our work in more detail.
> The more descriptions have been supplemented in Section 1, 5 and Appendix A.4.
>
> 2. Reviewer: How does BRMORL compare to RMORL or SRMORL?
>
> Author: Thank you for your valuable comments. In the current version, we have made further improvements. In Section 5 and Appendix A.4, we added the comparative tests based on RMORL, SRMOR. Please refer to Section 5 and Appendix A.4 for details.
>
> 3. Reviewer: It would help to have an algorithm box for BRMORL, that clarifies how the adversary policy, protagonist policy, and Bayesian optimization are used to gather data and for training.
>
> Author: Thank you for your valuable comments. In the current version, we have made further improvements. In Appendix A.1.1, the details of the BRMORL, RMORL, SRMORL and SMORL schemes are provided in Algorithm 1, 2, 3 and 4 respectively.
>
> 4. Reviewer: The proposed metric is questionable. The goal is to capture both diversity and quality of solutions, but in Figure 3, I would argue that Pareto front 1 is indeed better, because these points dominate all of the points on Pareto fronts 2 and 3, and the purpose of MORL is to find non-dominated policies.
>
> Author: Thank you for your valuable comments. In the current version, we have made further improvements. In Section 4.4, We added a more accurate and detailed descriptions.
> In fact, what we want to illustrate is similar to the situation in Figure 6(c). For the same problem, the Pareto front approximated by different algorithms is generally different, and the real Pareto front is also unknown. In Figure 4 (the current version), the Pareto frontiers 1, 2 and 3 are obtained by different algorithms, instead of same algorithm.
> From the perspective of the multi-objective optimization, Pareto front 1 is indeed better than the other two "Pareto fronts".
> In practice, however, we think that sometimes more effective options may be more important. For example, an electric car participating in a competition will pay more attention to speed rather than energy saving; energy conservation would be more important if the electric car had little residual power; in many cases, when driving this electric car, we have to make a tradeoff between speed and energy efficiency. Therefore, we hope that our trained model can approximate the "good"(robust, diverse, well-distributed, Pareto optimal or even suboptimal) policy for any specified preference.

---

### Official Review · AnonReviewer3 · 2020-10-28
**Interesting idea, but not ready for publication**

**Rating:** 3
**Confidence:** 5

**Review:**

The paper proposes a novel approach for solving MORL problems while considering uncertainty in the Pareto frontier.
The contributions are interesting and novel, but the paper has several flaws which make it not ready for acceptance, especially regarding experiments.

First, the authors overlook many Bayesian MORL algorithms which also consider the Pareto frontier uncertainty. For instance,

Calandra et al, "Pareto Front Modeling for Sensitivity Analysis in Multi-Objective Bayesian Optimization"

Calandra et al, "Bayesian Multiobjective Optimisation With Mixed Analytical and Black-Box Functions: Application to Tissue Engineering")

Hernandez-Lobato et al, "Predictive entropy search for multi-objective Bayesian optimization"

Olofsson et al, "Bayesian multi-objective optimisation of neotissue growth in a perfusion bioreactor set-up"

In the introduction the authors say "In addition, most approaches still only work in domains with low-dimensional and discrete action spaces." This is not true. Simply, all cited algorithms have just been tested on low-dimensional problems. Since they were not evaluated on larger problems, we do not know how they behave. The authors do not even include any of them in the evaluation to show that they actually fail.
Furthermore, the authors claim that their experiments have large action spaces. How big are they? This is not mentioned in the paper. For instance, the Mujoco environments tested in the paper do not have that large action spaces (eg, Swimmer has 15 actions).

The writing can be also improved. The sections feel a bit disconnected, and sometimes it is not easy to highlight the contributions. For instance, Section 4.3.1 takes quite some space and seem to be part of the novelty contributions, but the losses are taken from Yang et al.

My biggest issue is with the experiments. First, the authors should say right away on which environment they are testing the algorithms, and not just "Mujoco" and "two provided by Xu et al". Furthermore, these two environments are never actually tested, since the experiments are only on SUMO, Swimmer, Walker, and HalfCheetah.
And why are results for the Swimmer shown as figure, while Walker and Cheetah have tables? And why these environments out of all Mujoco ones? (these three are known to be the easiest).
Moreover, Mujoco environments are single-objective. How did you turn them into multi-objective? This is crucial and not mentioned.
Finally, there is no evaluation against any of the MORL algorithms mentioned in related work. The paper makes the point that these should fail with large action spaces and uncertainty, but this is not shown.

Overall, the experiments section feels rushed and incomplete, and the paper is not ready for publication.

** EDIT **
The authors added many experiments, tables, figures, sections, and an appendix. The changes are too substantial and the paper looks like a completely new one.
The purpose of author rebuttals is to address issues like a reviewer’s uncertainty about a point, an incorrect assumption, a misconception, or a misunderstanding of a part of the paper, not to revamp the paper completely.
The paper was clearly incomplete at the time of its submission, and I still vote for its rejection.

---

> ### Author Response · Authors · 2020-11-23
> **Answer the experts' questions and explain the revision of the paper**
>
> 6. Reviewer: Why are results for the Swimmer shown as figure, while Walker and Cheetah have tables.
>
> Author: Thank you for your valuable comments. In the current version, we have made further improvements, and the test results based on the Swimmer, Walker and Cheetah are shown in figures and tables. Please refer to the experimental section and appendix for details.
>
> 7. Reviewer: Why these environments out of all Mujoco ones? (these three are known to be the easiest).
>
> Author: Thank you for your valuable comments. First, Mujoco is widely used to verify the performence of reinforcement learning algorithm. Second, our work focuses on enabling agent to learn robust Pareto optimal policies rather than solving high-dimensional space problems, hence, Swimmer, Walker, HalfCheetah and SUMO can verify the robustness, diversity, uniformity, and Pareto optimality of the learned policies.
>
> 8. Reviewer: Mujoco environments are single-objective. How did you turn them into multi-objective?
>
> Author: Thank you for your valuable comments. For this problem, Please refer to appendix A.2 in the latest paper.
>
> 9. Reviewer: There is no evaluation against any of the MORL algorithms mentioned in related work. The paper makes the point that these should fail with large action spaces and uncertainty, but this is not shown.
>
> Author: Thank you for your valuable comments. We have made further improvements in the current version.
>
> In the experimental section, we compared the MORL based on weighted sum. Morever, we compare our BRMORL scheme with state-of-the-art algorithm provided by Xu et al. (2020). Please refer to the experimental section and appendix for details.

---

> ### Author Response · Authors · 2020-11-23
> **Answer the experts' questions and explain the revision of the paper**
>
> We appreciate very much for your time and valuable comments in reviewing our work. We have made a lot of improvements in the current version.
>
> 1. Reviewer: The authors overlook many Bayesian MORL algorithms which also consider the Pareto frontier uncertainty.
>
> Author: Our paper focuses on robust multi-objective reinforcement learning problem, and Bayesian optimization algorithm is adopted to guide the agent to evolve towards improving the quality of the approximated Pareto frontier. The four papers listed are all about multi-objective optimization, which is different from multi-objective reinforcement learning. The goal in reinforcement learning is to determine a policy which maximizes expected return when the agent acts according to it, which is sequential decision-making problem. Optimization problem can be defined as determining a solution that are within the feasible region to minimize (maximize) an objective function, which is a broader domain than reinforcement learning. Because this paper focuses on reinforcement learning problem and the paper has strict space limitation, we mainly introduce the researches of multi-objective reinforcement learning and robust reinforcement learning in Section 2 (Related work).
>
> Of course, if the listed literatures about Bayesian multi-objective optimization is introduced, our work will be better. But there is really no room for this paper, so we can only introduce the most directly related researches. Thank you for your valuable suggestions.
>
> 2. Reviewer: In the introduction the authors say "In addition, most approaches still only work in domains with low-dimensional and discrete action spaces." This is not true. Simply, all cited algorithms have just been tested on low-dimensional problems. Since they were not evaluated on larger problems, we do not know how they behave. The authors do not even include any of them in the evaluation to show that they actually fail. Furthermore, the authors claim that their experiments have large action spaces. How big are they? This is not mentioned in the paper. For instance, the Mujoco environments tested in the paper do not have that large action spaces (eg, Swimmer has 15 actions).
>
> Author: Thank you for your valuable comments. For this problem, we have made further improvements in the current version.
>
> "In addition, most approaches still focus on the domains with discrete action space. In contrast, our scheme can guarantee the learned policies is approximately robust Pareto-optimal on continuous control tasks."
>
> We propose a generalized robust MORL framework based on BO, but there is no specific innovation and research on low-dimensional problems or large action spaces.
>
> 3. Reviewer: The writing can be also improved. The sections feel a bit disconnected, and sometimes it is not easy to highlight the contributions. For instance, Section 4.3.1 takes quite some space and seem to be part of the novelty contributions, but the losses are taken from Yang et al.
>
> Author: Thank you for your valuable comments. We have made further improvements in the current version.
>
> In Section 4.3.1, except that the auxiliary loss setting framework is the same, the rest is completely different from Young's work, and we have added more detailed formula derivations and descriptions. The core difference is that our policy evaluation is carried out under the attack of adversary and combines the preference optimized by Bo algorithm.
>
> 4. Reviewer: The authors should say right away on which environment they are testing the algorithms, and not just "Mujoco" and "two provided by Xu et al".
>
> Author: Thank you for your valuable comments. We have made further improvements in the current version. In the experiment section and appendix, a large number of experimental and algorithmic details are supplemented.
>
> 5. Reviewer: These two environments are never actually tested, since the experiments are only on SUMO, Swimmer, Walker, and HalfCheetah.
>
> Author: Thank you for your valuable comments. The Walker and HalfCheetah are two MORL domains provided by Xu et al. The relevant descriptions have been supplemented in the experimental section and appendix.

---

### Public Comment · ~Jie_Xu7 · 2020-11-10
**Questions about the Pareto metric and the experiment**

Hi, I'm the author of PGMORL (Xu et al. 2020). The paper proposes an interesting approach to solve the MORL problem from the Bayesian Optimization perspective. I have two major questions regarding the claims on the Pareto metrics and the comparison results to our method.

First, in section 4.4, the paper claims the hypervolume and the sparsity metrics are not the ideal ones. To support the claim, the paper uses a simple illustration in Figure 3. However, this is not a good example to illustrate this claim. The ultimate goal of multi-objective optimization is to maximize $\vec{F}(x)$, which results in a set of Pareto-optimal solutions, ideally each of which cannot be dominated by other solutions. In Figure 3, from the perspective of the multi-objective optimization, Pareto front 1 is definitely a better solution set than the other two "Pareto fronts", since the points on the other two fronts are entirely dominated by the solutions in front 1. The small range of the Pareto front 1 potentially results from the problem itself where the objectives are not conflicting enough so that one single solution that can reach optimality on each objective exists, instead of the metrics. So I think a more rigorous explanation will be helpful.

Second, the results of the comparison experiments to PGMORL look super great, though the correctness of the experiments may need to be concerned. From the data reported for PGMORL, RA, PFA, META, RANDOM, I assume that the author uses the exactly same environments and problem settings as PGMORL (since those values are the same as PGMORL reported). For HalfCheetah-v2 problem, each objective does not exceed 5, and the total number of environment steps is 500, which means the best policy can only achieve the reward (2500, 2500). Thus in HalfCheetah-v2, our objective setting provides a theoretical upper-bound for the hypervolume ($6.25 \times 10^6$), which is much smaller than the hypervolume value ($12.76\times 10^6$) achieved by the proposed method. For the Walker2d-v2 problem, the first objective (running speed) has no upper-bound, but through single-objective RL (PPO) for the running objective, we get an empirical upper-bound for it as 2600, so it means there is an empirical upper-bound for the Walker2d-v2 problem as $2600 \times 2500 = 6.5 \times 10^6$, while the reported hypervolume in the paper is $8.84 \times 10^6$.  Since we change some physics parameters (i.e. mass, friction, etc) in mujoco walker environment to make the objectives more conflicting to each other, I'm wondering whether the author uses the same environment setting and the calculation of the objective for a fair comparison or the improvement on the Walker problem results from the difference of the underlining RL algorithm?. For the details of the objective setting, please refer to our supplementary material: http://people.csail.mit.edu/jiex/papers/PGMORL/supp.pdf. A more detailed experiment description regarding to that part is appreciated.

---

> ### Author Response · Authors · 2020-11-13
> **Respond to experts’ questions and clarify the views of our work.**
>
> Hi, the author of PGMORL, thank you for your valuable comments.
>
> For first question, maybe we didn't explain it clearly, so it caused a lot of misunderstanding. In fact, what we want to illustrate is similar to the situation in Figure 5 (c). For the same problem, the Pareto front approximated by different algorithms is generally different, and the real Pareto front is also unknown. In Figure 3, the Pareto frontiers 1, 2 and 3 are obtained by different algorithms, instead of same algorithm. From the perspective of the multi-objective optimization, Pareto front 1 is indeed better than the other two "Pareto fronts". In practice, however, we think that sometimes more effective options may be more important. For example, an electric car participating in a competition will pay more attention to speed rather than energy saving; energy conservation would be more important if the electric car had little residual power; in many cases, when driving this electric car, we have to make a tradeoff between speed and energy efficiency. Therefore, we hope that our trained model can approximate the "good"(robust, diverse, well-distributed, Pareto optimal or even suboptimal) policy for any specified preference. Although the small range of Pareto frontier 1 may be caused by the problem itself, we can increase the diversity of solutions through the appropriate method. We will improve this part of the description recently to avoid misleading readers' understanding. Thanks again for your valuable suggestions.
>
> For second question, we think that the results of the experiment are different mainly due to different algorithms. For an on-policy RL algorithm(e.g. PPO), the method you calculate theoretical upper-bound for the hypervolume should be no problem. However, for an off-policy RL algorithm, the collected transition data can be reused, so the theoretical upper-bound for the hypervolume is not equal to (maximum reward × environment steps)². Using different RL algorithms may make the results different. We have tried to use the SAC algorithm to implement our BRMORL scheme, but the effect is not as good as the current DDPG scheme (does not rule out the possibility that we did not adequately debug the scheme based on the SAC algorithm). Therefore, from the perspective of RL algorithm implementation, the scheme based on off-policy can have a higher theoretical upper-bound for the hypervolume then the scheme based on on-policy. We use your the "environments" folder and the "environment.yml" file to build Walker and HalfCheetah training environment, and we are adding more details to the paper recently.
>
> Our views may conflict with the views of your paper (a meta policy is a general policy that is not necessarily optimal) to some extent. Our view is to satisfy that the policy is robust, diverse and well-distributed, sometimes it is acceptable that the policy is not Pareto optimal. Although our approach cannot guarantee the learned policies are optimal, it is approximately robust Pareto optimal. This is like classical robust control, in order to ensure that the control system is robust to perturbation, control system need to sacrifice part of the performance to obtain the robust optimal solution.
>
> Recently we are working hard to improve our paper based on the reviewers' comments, if you have any suggestions, please feel free to comment. Thank you very much again.

---

> > ### Public Comment · ~Jie_Xu7 · 2020-11-13
> > **Response**
> >
> > Thanks for the response and for the clarification. I think overall this is a good submission and give us an interesting and effective approach to improve the existing works. I am still confused about the hypervolume computation for the results. To evaluate the hypervolume of the solution set (the real performance), we need to take all the policies on the Pareto front, run each of them for a fixed number of iterations (500 in our case) to compute their performance on two objectives individually, then compute the hypervolume of them. In this evaluation stage, no RL algorithm gets involved, thus the theoretical upper bound for halfcheetah should be satisfied no matter which underline RL algorithm is used.  I think a better explanation of the experiment and a fair comparison will be helpful.

---

> > > ### Author Response · Authors · 2020-11-18
> > > **Answer experts' questions**
> > >
> > > Hi, Dr. Xu, thank you for your valuable suggestions.
> > >
> > > This time we understand how hypervolume is evaluated in your paper,  and our way to evaluate hypervolume is indeed somewhat different. Considering the training efficiency, we estimate the hypervolume based on the Q value using the last batch of data collected during each episode of training. Therefore, we may refer to your evaluation method and conduct a fair comparison test.
> > >
> > > Our revised paper is about to be completed, if you have any suggestions, please feel free to comment.
> > >
> > > Thanks again for your valuable comments.

---

### Author Response · Authors · 2020-11-25
**Overview for revision**

We appreciate the experts for your constructive comments and suggestions which are very insightful and thoughtful.

We have made a lot of improvements in the latest version. We added a large number of the details for algorithms and experiments, added more detailed supplementary tests, and optimized models, and so on. Moreover, Table 3 (the original version) has been deleted, because the work of Xu et al. (2020) has compared those baseline methods (RA, PFA, MOEA/D, RANDOM, META), we only need to compare the state-of-the-art scheme (PGMORL) provided by Xu et al. (2020).

We have studied the reviewers’ comments carefully, and a thorough revision has been made accordingly (marked in yellow in the paper).



PS: Our view is to satisfy that the policy is robust, diverse and well-distributed, sometimes it is acceptable that the policy is not Pareto optimal. Although our approach cannot guarantee the learned policy is optimal, it is approximately robust Pareto optimal. This is like classical robust control, in order to ensure that the control system is robust to perturbation, control system need to sacrifice part of the performance to obtain the robust optimal solution.

---

### Decision · Program_Chairs · 2021-01-07
**Final Decision**

**Decision:**

Reject

**Comment:**

The paper studied multi-objective reinforcement learning (MORL), and provided a Bayesian optimization approach for challenging MORL scenarios in several simulation environments. The reviewers generally find it interesting to account for robustness in a MORL setup, and all appreciate the algorithmic contributions. However, there were shared critical concerns among \ reviewers in the technical clarity and positioning of the work.

The paper has gone through substantial changes during the rebuttal period, which addressed some concerns regarding the experimental details; however, the major revision raised further issues that affects the clarity of the work. The reviewers are hence unconvinced that the paper is ready for publication. In addition to addressing the existing comments on clarifying the experimental details and properly positioning the work against prior art, a reorganization and optimization of the main content would be beneficial for future submission.